# Bête Noire of Chemotherapy and Targeted Therapy: CAF-Mediated Resistance

**DOI:** 10.3390/cancers14061519

**Published:** 2022-03-16

**Authors:** Pradip De, Jennifer Aske, Raed Sulaiman, Nandini Dey

**Affiliations:** 1Translational Oncology Laboratory, Avera Cancer Institute, Sioux Falls, SD 57105, USA; pradip.de@avera.org (P.D.); jennifer.aske@avera.org (J.A.); 2Department of Pathology, Avera McKennan Hospital, Sioux Falls, SD 57105, USA; raed.sulaiman@plpath.org

**Keywords:** cancer-associated fibroblasts, resistance, chemotherapy, targeted therapy

## Abstract

**Simple Summary:**

Tumor cells struggle to survive following treatment. The struggle ends in either of two ways. The drug combination used for the treatment blocks the proliferation of tumor cells and initiates apoptosis of cells, which is a win for the patient, or tumor cells resist the effect of the drug combination used for the treatment and continue to evade the effect of anti-tumor drugs, which is a *bête noire* of therapy. Cancer-associated fibroblasts are the most abundant non-transformed element of the microenvironment in solid tumors. Tumor cells play a direct role in establishing the cancer-associated fibroblasts’ population in its microenvironment. Since cancer-associated fibroblasts are activated by tumor cells, cancer-associated fibroblasts show unconditional servitude to tumor cells in their effort to resist treatment. Thus, cancer-associated fibroblasts, as the critical or indispensable component of resistance to the treatment, are one of the most logical targets within tumors that eventually progress despite therapy. We evaluate the participatory role of cancer-associated fibroblasts in the development of drug resistance in solid tumors. In the future, we will establish the specific mode of action of cancer-associated fibroblasts in solid tumors, paving the way for cancer-associated-fibroblast-inclusive personalized therapy.

**Abstract:**

In tumor cells’ struggle for survival following therapy, they resist treatment. Resistance to therapy is the outcome of well-planned, highly efficient adaptive strategies initiated and utilized by these transformed tumor cells. Cancer cells undergo several reprogramming events towards adapting this opportunistic behavior, leading them to gain specific survival advantages. The strategy involves changes within the transformed tumors cells as well as in their neighboring non-transformed extra-tumoral support system, the tumor microenvironment (TME). Cancer-Associated Fibroblasts (CAFs) are one of the components of the TME that is used by tumor cells to achieve resistance to therapy. CAFs are diverse in origin and are the most abundant non-transformed element of the microenvironment in solid tumors. Cells of an established tumor initially play a direct role in the establishment of the CAF population for its own microenvironment. Like their origin, CAFs are also diverse in their functions in catering to the pro-tumor microenvironment. Once instituted, CAFs interact in unison with both tumor cells and all other components of the TME towards the progression of the disease and the worst outcome. One of the many functions of CAFs in influencing the outcome of the disease is their participation in the development of resistance to treatment. CAFs resist therapy in solid tumors. A tumor–CAF relationship is initiated by tumor cells to exploit host stroma in favor of tumor progression. CAFs in concert with tumor cells and other components of the TME are abettors of resistance to treatment. Thus, this liaison between CAFs and tumor cells is a *bête noire* of therapy. Here, we portray a comprehensive picture of the modes and functions of CAFs in conjunction with their role in orchestrating the development of resistance to different chemotherapies and targeted therapies in solid tumors. We investigate the various functions of CAFs in various solid tumors in light of their dialogue with tumor cells and the two components of the TME, the immune component, and the vascular component. Acknowledgment of the irrefutable role of CAFs in the development of treatment resistance will impact our future strategies and ability to design improved therapies inclusive of CAFs. Finally, we discuss the future implications of this understanding from a therapeutic standpoint and in light of currently ongoing and completed CAF-based NIH clinical trials.

## 1. Introduction

Cancer-associated fibroblasts (CAFs) within the tumor microenvironment (TME) are non-transformed, tumor-cell-activated heterogeneous populations of cells having multiple origins and functions [1,2]. Detailed descriptions of the origin, functions, interactions with tumor cells, and heterogeneity of CAFs were previously provided by us elsewhere [3,4]. CAFs are activated by tumor cells in their favor. Once activated in an established tumor, CAFs act as crucial supporters of tumor growth, progression, and response to treatment.

The functions of CAFs in an established tumor include the following: (1) ECM (extracellular matrix) remodeling via collagenolysis to promote invasion and EMT (endothelial–mesenchymal transition); (2) increasing tissue stiffness to initiate angiogenic resistance and immune suppression; (3) induction of tumor angiogenesis; (4) secretomic induction of EMT by TGFbeta; (5) increasing secretomic factors of tumor-promoting or immune-suppressing ligands such as hepatocyte growth factor; fibroblast growth factors 1 and 2; stromal cell-derived factor 1 (SDF1/CXCL12); chemokine (C-C motif) ligands (CCL) 2, 5, 7, and 16; interleukin 6/8; and platelet derived growth factor; (6) metabolic reversal of reverse Warburg effect (non-glycolysis in tumor cells, glycolysis in stroma cells) and ‘lactate shuttle’ effect; (7) immune evasion via activation of M2 macrophages (CD163 positive); (8) inhibition of apoptosis in tumor cells; (9) activation of many of pro-proliferative tumor cell signaling; (10) immune reprogramming and antigen presentation; (11) adaptation to oxidative stress and hypoxic response; (12) promotion of stemness-promoting signals; (13) promotion of metastasis-associated phenotypes; (14) attenuation of drug response [1,5,6,7,8,9,10,11,12,13,14,15,16,17,18].

The range of functions of CAFs is comprehensive, and the actions of CAFs are contextual. The interactions of CAFs with tumor cells and TME components change with the evolution of the tumor, its metastatic progression, and its response to therapy. In summary, the functions of CAFs are structured to assist and promote tumor cells via direct and indirect interactions. Thus, CAFs form a centralized communication network within the TME that favors tumor cell growth, metastasis, and resistance to drug treatment [19]. The versatility of the functions of CAFs’ make them abettors of drug resistance and identifies them as prospective anti-tumor therapy targets [20,21]. Here, we investigate the role of CAFs in the development of resistance to chemotherapy and targeted therapy. We seek to evaluate whether co-targeting CAFs will have a participatory benefit towards managing the burden of resistance. We discuss the opportunity that CAFs present to improve and evolve the management of the disease from a tumor-centric approach to a *tumor–CAF-centric approach*.

## 2. CAF Heterogeneity and Resistance to Chemotherapy in Solid Tumors

### 2.1. CAF Heterogeneity

CAFs are heterogeneous in terms of their origin in different organ-type cancers, as well as in the progression of the disease. The heterogeneous subpopulations of CAFs, such as myoblastic CAFs (myCAFs) and inflammatory CAFs (iCAFs), have been extensively studied in fibroinflammatory PDAC disease characterized by dense and highly proliferating desmoplastic stroma. In fact, Li et al. identified genes associated with the differentiation of myCAFs and iCAFs [22,23,24]. Adipose-derived MSCs (AD-MSCs) have been shown to possess a high multilineage potential and self-renewal capacity and were reported as the CAF sources in PDAC by Miyazaki et al. [24]. Their study identified that AD-MSCs could differentiate into distinct CAF subtypes, myCAFs and iCAFs, depending on the different co-culture conditions in vitro. The diverse functions of iCAFs and myCAFs have also been reported in cholangiocarcinoma; breast cancers; prostate, head, and neck squamous cell carcinoma; and bladder and colon cancers. The diversity of CAF subpopulations was also recently reported to promote the growth of cholangiocarcinoma, wherein hepatic stellate cells (HSC) are the primary cause of CAF differentiation into myCAFs and iCAFs [25]. The hyaluronan synthase 2 myCAFs, but not type I collagen-expressing myCAFs, promoted tumor progression, while HGF-expressing iCAFs enhanced tumor growth via tumor-expressed MET, thereby directly linking CAFs to tumor cells. Another subset of CAFs, FAP+CAFs, were identified by Kieffer et al. in breast cancers that mediated immunosuppression and immunotherapy resistance via a positive feedback loop between specific CAF-S1 clusters and Tregs [26]. In prostate cancer, a differential mode of activation of iCAFs and myCAFs has been reported [27]. IL-1a/ELF3/YAP pathways are involved in iCAF differentiation, while TGF-beta1 induces myCAFs. One of the ways CAFs classically interact with the tumor cell EMT function was reported by Goulet et al. in bladder cancer, where IL-6 cytokine was found to be highly expressed in iCAFs, and its receptor IL-6R was found on RT4 bladder cancer cells [28]. Perhaps the most intriguing functional heterogeneity of CAFs was reported by Pan et al. in PDAC-CAF-exhibited organ-specific metastatic potential leading to different levels of heterogeneity of CAFs in different metastatic niches [29]. Several cell signaling pathways have been reported to be involved in the functioning of iCAFs and myCAFs, including the Hedgehog pathway [30]; Wnt pathway [31]; integrin a11B1 signaling [32]; cMET-HGF pathway [25]; IL-6 signaling [28]; EMT signaling via transcription factors SNAIL1, TWIST1, and ZEB1 [28]; and IL1B-mediated crosstalk [33]. Recently, Steele et al. reported that the Hedgehog pathway acts in a paracrine manner in PDAC, with ligands secreted by tumor cells signaling to stromal CAFs. The Hedgehog pathway activation is higher in PDPN+ alphaSMA+ myCAFs compared with iCAFs, and its inhibition impairs tumor growth by altering the fibroblast compartment in PDAC. Hedgehog pathway inhibition resulted in a reduction in myCAF numbers and a significant expansion of iCAFs, leading to an increase in the iCAF/myCAF ratio. As iCAFs are a source of inflammatory signals, the authors observed an increase in iCAFs upon Hedgehog inhibition, which correlated with changes in immune infiltration (significantly decreased CD8+ T cells and increased CD4+ T cells and CD25+CD4+ T cells; abundant FOXP3+ regulatory T cells) that are consistent with a more immunosuppressive pancreatic cancer microenvironment. The paracrine activation differentially elevated myCAFs compared with iCAFs, leading to favorable alterations of cytotoxic T cells and Tregs, causing increased immunosuppression [30]. Wnt signaling in CAFs represents a non-cell-autonomous mechanism for colon cancer progression [31]. Mose et al. reported Sfrp1 epithelial–mesenchymal transition phenotype induction in tumor cells without affecting tumor-intrinsic Wnt signaling, suggesting involvement of non-immune stromal cells. Low levels of Wnt signals induced the iCAF subtype, which in co-culture with organoids induced EMT, whereas high levels induced contractile myCAFs to attenuate the EMT phenotype.

The tumors with (1) an accumulation of stromal CAFs, (2) the presence of fibrotic stroma, (3) a high expression level of stroma signature genes, or (4) a high tumor/stroma ratio in the primary tumor are associated with poor prognosis in various cancers, including colon, gastric, esophagus, breast, NSCLC (non-small cell lung cancer), and liver cancers [34,35,36,37,38,39,40]. It is understood that chemotherapy’s limited effect (benefit) and the progression or recurrence of disease through therapy in many solid tumors are attributed to the development of resistance within tumor cells in support of the stroma. As a dominant component of tumor stroma, CAFs interact with both a tumor cell and the TME. The versatility of CAF functions and their several modes of interaction with tumor cells and all components of stroma (ECM and cells of the TME) indicate that a metastasis or progression of disease following treatment is aided and abetted by CAFs. Once a therapy-resistive circuitry is established between tumor cells and the CAFs of the stroma, tumor-centric therapy alone essentially becomes insufficient. Figure 1 presents the distribution pattern of the types of resistance to chemotherapy based on specific mediators of CAF functions in solid tumors. The four types of mediators of action employed by CAFs to orchestrate the development of resistance to chemotherapy are presented in the cartoon. The most common mode of interaction is a paracrine, wherein CAFs signal to either tumor cells or other components of the TME via characteristic secretomes. In addition to the involvement of characteristic secretomes, exosomal cargos delivering different miRNAs that target various cell signaling proteins are common mediators of CAFs. The paracrine mode of action of CAFs is the predominant form of action, represented by six types of organ tumors (organ tumors are indicated by their respective ribbon colors, as presented in the figure legends). CAF crosstalk with tumor cells, and the TME occurs via exosomal cargo, imparting resistance to four organ cancers. The extracellular vesicle, secretome, and autocrine or paracrine modes are much less involved in the modes of action (Figure 1). The sizes of the boxes indicate the number of studies in each box. Among resistance to different types of chemotherapies, cisplatin resistance has been found to be very common, which is involved in both paracrine and exosomal cargo modes of action (the shapes in the inset indicate the types of resistances in different tumors).

Among the different types of solid tumors, gastric cancers have been reported to be the most common tumors exhibiting CAF-mediated resistance to chemotherapy, which involve paracrine, exosomal cargo, extracellular vesicle, and secretomic modes of action. Secretion of IL-11 from CAFs activated the IL-11/IL-11R/gp130/JAK/STAT3/Bcl anti-apoptosis signaling pathway in gastric cancer cells. Thus, CAF-derived IL-11 secretion caused resistance to chemotherapy regimens in gastric cancers [41]. In another study, CAF-induced activation of the JAK-STAT signaling has been proposed to confer chemoresistance in gastric cancer cells, while interleukin-6 (IL-6) was identified as a CAF-specific secretory protein that protects gastric cancer cells via paracrine signaling. Interestingly, clinical data have shown that IL-6 was differentially expressed in the stromal portion of cancer tissues, while IL-6 upregulation was positively correlated with poor responsiveness to chemotherapy [42]. In line with the above facts, several CAF-targeting agents have been tested in experimental models, as reviewed elsewhere [43]. Resistance to conventional chemotherapeutics in gastric cancers has been reported to be mediated by CAF-derived extracellular vesicles [44]. Annexin A6 initiated network formation and drug resistance within the ECM via activation of beta1 integrin-FAK-YAP signaling. Annexin A6 within CAF extracellular vesicles has been shown to stimulate FAK-YAP signaling by stabilizing beta1 integrin at the cell surface of gastric cancer cells, which subsequently induces drug resistance. In addition to extracellular vesicles, CAFs also communicate via exosomal cargos, which carry miRNAs and mediate resistance to specific chemotherapeutic agents, as presented in the following section.

### 2.2. CAFs and Specific Resistance to Cisplatin

Reports of CAF-mediated development of cisplatin resistance are more prevalent than any other chemotherapy agent. In certain solid tumors, the mechanism involved intracellular pathway signaling such as JNK or NF-κB, adhesion molecules such as annexin A3, or specific proteins such as plasminogen activator inhibitor-1. In lung cancers, CAFs have been reported to express a higher level of annexin A3 (ANXA3) than normal fibroblasts. The crosstalk was demonstrated using CAF-CM (CAF-conditioned media) incubation, which increased the ANXA3 level in lung cancer cells, which subsequently enhanced cisplatin resistance by inhibiting cisplatin-induced apoptosis involving ANXA3/JNK signaling [45]. In lung adenocarcinoma, cisplatin resistance was associated with the expression of SMAalpha expression [46]. In their study, Masuda et al. demonstrated that the inhibition of plasminogen activator inhibitor-1 increased the chemotherapeutic effect in lung cancer through suppressing the myofibroblast characteristics of CAFs. CAF-derived IL-8 promoted chemoresistance to cisplatin in gastric cancer via NF-κB activation and ABCB1 upregulation [47]. In bladder cancers, stromal CAFs enhanced cisplatin resistance via stimulating IGF-1/ERbeta/Bcl-2 signaling, wherein CAFs regulated ERbeta expression through IGF-1/AKT/c-Jun signaling following c-Jun phosphorylation and promoted ESR2 gene transcription [48]. In other cancers, exosomal cargo carried miRNA to mediate the CAFs’ effect. In ovarian cancer, CAF-mediated cisplatin resistance was reported to involve CAF-derived exosomes, which overexpressed miR-98-5p [49]. In immunocompromised mice, miR-98-5p targeted CDKN1A to inhibit CDKN1A expression and promoted cisplatin resistance by virtue of cell cycle progression. In head and neck cancer, cisplatin resistance is perpetrated by CAF-derived exosomal miR-196a targeting CDKN1B and ING6 [50]. Whether the nature of CAF mediators of cisplatin resistance is organ-specific or not needs to be concluded with more data in this field. From the current literature, it is evident that exosomal miRNA predominantly mediates platinum-based chemotherapy resistance (cisplatin and oxaliplatin), with a few exceptions such as tamoxifen resistance in breast [51] and radioresistance in colorectal cancers [52,53]. In the context of resistance to radiotherapy, CAFs are highly radio-resistant, even at high doses of radiation. CAFs resist apoptosis signals following radiation and become senescent, producing a distinct combination of immunoregulatory molecules. Hence, acquired radio resistance has been associated with CAF function [54,55]. A recent minireview summarized findings on the interactions between CAF, ionizing radiation, and immune cells in the tumor microenvironment [56]. Targeting CAFs, regulatory T cells, and tumor-associated macrophages in combination radio–immunotherapies has been reported to improve cancer treatment [57]. Future studies will also need to clarify the functional segregation of the two modes of events and whether it exists in the development of CAF-mediated resistance in solid tumors.

### 2.3. CAFs and Specific Resistance to Paclitaxel

CAF-mediated resistance to paclitaxel was reported in ovarian cancers. In ovarian cancers, the lipoma-preferred partner gene has been reported to mediate CAF–endothelial cell crosstalk in signaling chemoresistance [58]. CAFs upregulated the lipoma-preferred partner gene in microvascular endothelial cells via calcium-dependent signaling, and lipoma-preferred partner expression levels in intratumoral microvascular endothelial cells correlated with survival and chemoresistance in patients. Lipoma-preferred partners upregulated focal adhesion and stress fiber formation to promote endothelial cell motility and permeability. Experimental suppression of lipoma-preferred partners improved paclitaxel delivery to cancer cells by decreasing intratumoral microvessel leakiness.

### 2.4. CAFs and Specific Resistance to a Combination of Cisplatin and Paclitaxel

Specific resistance to a combination of cisplatin and paclitaxel aided by CAFs is encountered in gastric cancers. Exosomal miR-522 suppressed ferroptosis and promoted acquired chemoresistance (decreased chemosensitivity) by targeting ALOX15 and blocking lipid–ROS accumulation involving the intercellular pathway. Both cisplatin and paclitaxel treatment promoted miR-522 secretion from CAFs by activating the USP7/hnRNPA1 axis, leading to ALOX15 suppression and decreased lipid–ROS accumulation in gastric cancer cells [59].

### 2.5. CAFs and Specific Resistance to Oxaliplatin

CAFs orchestrate oxaliplatin resistance in colorectal cancers [60]. Colorectal cancer-associated lncRNA is transferred from CAFs to the cancer cells via exosomes, where it suppresses colorectal cancer (CRC) cell apoptosis, confers chemoresistance, and activates the Wnt/beta-catenin pathway. Long-non-coding RNA interacts directly with mRNA stabilizing protein (human antigen R) to increase beta-catenin mRNA and protein levels. Specific resistance to 5-FU/L-OHP (oxaliplatin) has been reported in colorectal cancers. In colorectal cancers, chemotherapy resistance was attributed to CAF-secreted exosomes [61]. A direct transfer of exosomes to colorectal tumor cells led to a significant increase in miR-92a-3p levels in cancer cells. An increased expression of miR-92a-3p activated the Wnt/beta-catenin pathway and inhibited mitochondrial apoptosis by directly inhibiting FBXW7 and MOAP1, contributing to stemness, EMT, metastasis, and 5-FU/L-OHP resistance.

### 2.6. CAFs and Specific Resistance to Gemcitabine

CAF-mediated resistance to gemcitabine involves CAF-derived SDF-1. SDF-1 stimulated malignant progression and gemcitabine resistance in pancreatic cancer due to paracrine induction of SATB-1 within tumor cells. SDF-1-mediated upregulation of SATB-1 expression in tumor cells contributed to the maintenance of CAF properties, forming a reciprocal feedback loop involving the SDF-1/SATB-1 pathway [62]. It is apparent from the results of the above studies that mediators of CAFs in the development of resistance to different chemotherapeutics are specific not only to organ cancers but also the particular drug. In an ideal world, we should be searching for an organ-specific blood-based marker that can correlate or indicate CAF-mediated development of resistance to chemotherapy.

## 3. CAFs and Resistance to Targeted Therapy in Solid Tumors

CAF-mediated resistance to targeted therapy in solid tumors can be categorized into (1) specific resistance to hormone-receptor-targeted anti-cancer drugs and (2) specific resistance to non-hormonal pathway-targeted anti-cancer drugs (Figure 2). One characteristic feature of this type of resistance is the lack of mediation via miRNA compared to resistance to chemotherapy. The only exception to this characteristic is a novel subset of CD63+ CAFs that mediated resistance to tamoxifen in breast cancers via exosomal miR-22 [51]. CD63+ CAFs have been reported to secrete miR-22-rich exosomes, which act through its targets, ERalpha and PTEN, to confer tamoxifen resistance in breast cancer cells. The details of the development of resistance to hormone receptor-targeted anti-cancer drugs mediated by CAFs in breast cancers have been reviewed elsewhere [63]. CAFs have been involved in mediating anti-androgen resistance in prostate cancers in a paracrine manner. Zhang et al. identified neuregulin 1 (NRG1) in the CAF supernatant [64]. CAF-derived NRG1 promoted resistance in tumor cells through the activation of HER3 involving the NRG1/HER3 axis, proving a paracrine mechanism of anti-androgen resistance in prostate cancer. In line with the above fact, an inadequate response to second-generation anti-androgen therapy was recorded in castration-resistant patients with NRG1 activity.

The role of the activation of EGFR, Wnt/beta-catenin, Hippo, TGF-beta, and JAK/STAT cascades in CAFs in relation to the chemoresistance and invasive or metastatic behavior of cancer cells [65] has strengthened the concept that CAFs should be included as a target for therapy in solid tumors. CAF-mediated resistance to non-hormonal pathway-targeted anti-cancer drugs has been observed in lung, breast, melanoma, and hepatocellular cancers. CAF-mediated non-cell-autonomous adaptive resistance to MET- and EGFR-targeted therapies in lung cancers via a metabolic shift involving paracrine crosstalk between tumor cells under drug exposure and their surrounding CAFs has been reported [66]. Apicella et al. demonstrated that with prolonged exposure to tyrosine kinase inhibitors (TKIs), EGFR- or MET-addicted cancer cells undergo a metabolic shift upregulating glycolysis and lactate production. High secreted levels of lactate stimulate CAFs to produce hepatocyte growth factor (HGF) in a nuclear factor kappa B (NFkB)-dependent manner. This HGF, in turn, activates MET-dependent signaling within cancer cells, counteracting the effects of tyrosine kinase inhibitors (TKIs). In tumor cells of lung adenocarcinoma with EGFR mutations, primary EGFR-TKI resistance was associated with high hepatocyte growth factor in CAFs [67]. Conditioned media from CAFs increased the resistance of PC-9 cells to EGFR-TKI, indicating that with the secretion of higher amounts of CAF-derived humoral factors, HGF is responsible for EGFR-TKI resistance [67]. Understandably, this kind of fail-safe metabolic reprogramming not only allows cellular resistance to the drug but also re-establishes a tumor–TME circuitry, which can also merge with the local immune signaling [68,69,70,71]. As with prostate cancers [64] and melanomas [72], CAFs have been involved in developing resistance to targeted therapies in breast cancers. CAFs participate in the HER2-targeted therapy resistance in breast cancers via the TAF/FGF5/FGFR2/c-Src/HER2 axis [73]. CAF-derived NRG1 (an HER3 ligand) causes resistance to trastuzumab [74,75], TKIs [76], and T-DM1 [77] in HER2-positive breast cancers. In the Neosphere trial, HER2-positive breast tumors with high NRG1 expression appeared to resist trastuzumab–docetaxel but not pertuzumab–trastuzumab–docetaxel [78]. Guardia et al. identified CAFs as the primary source of NRG1 in HER2-positive breast cancers. The study showed their role in mediating resistance to trastuzumab, which can be overcome by dual anti-HER2 blockade following pertuzumab–trastuzumab [78]. Recently, a study examined the value of ‘pathological reactive stroma’ (defined as stromal-predominant breast cancer) as a predictor for trastuzumab resistance in patients with early HER2-positive breast cancer receiving adjuvant therapy in the FinHER phase III trial, reporting an association between trastuzumab resistance and the presence of ‘reactive stroma’ [79]. The pathological reactive stroma and the mRNA gene signatures that reflected reactive stroma were tested in 209 HER2-positive breast cancer samples and were found to be correlated with distant disease-free survival. Interestingly, reactive stroma did not correlate with tumor-infiltrating lymphocytes. The study concluded that the ‘pathological reactive stroma’ in HER2-positive or ER-negative early breast cancer tumors might predict resistance to adjuvant trastuzumab therapy.

In line with the pro-tumorigenic role of ‘pathological reactive stroma’, CAFs are known to promote organoid tumor growth in co-culture. The paracrine crosstalk between CAFs and cancer cells regulated physiological characteristics of CAFs, which in turn imparted resistance to cancer cells. In metastatic melanomas, CAFs resist the function of BRAF inhibitors via their crosstalk with tumor cells (vascular mimicry), the ECM, and endothelial cells (neovascularization). The development of drug resistance to BRAF inhibitors is mediated via ECM reprogramming action of CAFs [19]. Recently, Liu et al. reported the activation of nuclear beta-catenin signaling in melanoma CAFs during the development of resistance to BRAF inhibitor or MEK inhibitors, underscoring the role of BRAF-inhibitor-induced CAF reprogramming in matrix remodeling and the therapeutic escape of melanoma cells [80].

CAF populations expressing FAP/ITGA11/COL1A1/CCN2 have been shown to be negatively correlated with disease-free survival in this cancer. The resistance to BRAF inhibitors is the result of CAF-mediated reprogramming of the ECM. The stiffness of the ECM caused by CAFs has been associated with integrin-dependent signaling. Fibroblast-specific production of CCN2, whose overexpression in melanomas was independent of BRAF mutational status, signals through integrins and was found to be essential for neovascularization and vasculogenic mimicry. In hepatocellular carcinomas, tumor cells resist targeted anti-cancer drugs including sorafenib, regorafenib, and 5-fluorouracil in the presence of CAFs via a direct cell–cell contact, as tested in a transwell system through paracrine signaling [81].

CAF signaling in the development of drug resistance is tumor-specific in prostate cancers and lung adenocarcinomas, as presented above. In prostate cancers, CAF-derived neuregulin 1 NRG1 promotes resistance in tumor cells by activating HER3 involving the NRG1/HER3 axis, proving a paracrine mechanism of antiandrogen resistance in a paracrine manner, as presented above [64]. In lung adenocarcinomas bearing EGFR mutations, primary EGFR-TKI resistance is mediated via hepatocyte growth factor from CAFs. CM from CAFs increased the resistance of EGFR mutant lung adenocarcinoma cell line PC-9 cells to EGFR-TKI, indicating that the secretion of higher amounts of HGF is the robust feature of EGFR-TKI-resistance-promoting CAFs [67]. The mode of action of CAFs and the nature of their involvement with respect to the tumor cells and the TME are less studied. The pattern of crosstalk is just beginning to emerge, which can define distinct therapeutic paradigms. In a recent study, Engelman’s group reported three subtypes of lung CAFs that can influence the personalized treatment of non-small cell lung cancer patients. The 3 subtypes of CAFs identified in their study are (1) subtype I with HGF^High^, FGF7^High/Low^, p-SMAD2^Low^, targeting driver, HGF-MET, and FGF7-FGFR2; (2) subtype II with HGF^High^, FGF7^High^, p-SMAD2^Low^, targeting driver, and FGF7-FGFR2; and (3) subtype III with HGF^Low^, FGF7^Low^, and p-SMAD2^High^ [82]. They reported that specific subtypes are associated with particular functions and clinical responses. Subtype I and II CAFs function to protect cancer cells, while subtype III CAFs are involved with a better clinical response via immune cell migration with additional value in immuno-oncology. In addressing the heterogeneity of CAFs, the study systematically connected functions of subpopulations of lung CAFs to specific functions of CAFs in the context of clinical response and resistance to pathway-targeted drugs. Similar studies in the future will delineate the relationships of the mode of action of CAFs with drugs in organ-type cancers in solid tumors. Despite the different mediating actions of CAFs, it will be imperative to know how CAFs support a tumorigenic pathway in cancer cells in the face of pathway-targeted treatment that ultimately leads to the ineffectiveness of the therapy. Supplemental targeting of CAF signals opens an opportunity to improve personalized medicine and bears the promise of a better outcome.

## 4. Regulation of CAF Functions and Therapeutic Opportunity

### 4.1. CAFs as the Target within the TME

The irrefutable involvement of CAFs in the development of resistance to chemo- and targeted therapy and progression as presented above justifies the recognition of CAFs as a logical target for treatment. The interest in CAFs as a target of therapy arose from analyses of data from the conventional tumor-cell-centric view of cancer, targeting only the tumor component. The limited success of tumor-cell-centric therapies is a direct proof-of-concept that the TME bears undeniable responsibility for successful disease progression in solid tumors. From the conceptual aspect, any sequence-based therapy primarily refers to sequencing of the entire tumor tissue, which constitutes both cancer and the TME (CAFs along with immune cells and angiogenic components). Hence, the approach does not provide separate information on the subgroups, tumor cell cluster, CAF cluster, or immune cluster. Intratumoral heterogeneity contributes to the development of resistance to anti-cancer therapeutics. Thus, the heterogeneity of CAFs presents opportunities for CAF-targeted cancer therapies in precision medicine [65,83]. However, the burden of cost and management needs to be taken into account. CAFs as components of the TME have been targeted to suppress tumor growth [84]. Based on their specific surface markers and secreted molecules, Laplagne et al. reviewed the potential of targeting different aspects of CAFs, including cells inducing depletion, reprogramming, differentiation, or inhibition of their pro-tumor functions or recruitment. Several approaches involving immunotherapies, vaccines, small interfering RNA, or small molecules were developed to target components of the TME, as reviewed elsewhere [84].

CAFs are a coherent target in the TME [85]. The versatility of CAFs means they are a target for anti-tumor therapy to ‘switch off’ the pro-tumor stroma [20,65,86]. There are five ways to counter the CAF-mediated patronage of cancer cells, which eventually cause resistance to treatment and disrupt disease management. The strategic points to control the function of CAFs are (1) preventing the activation of CAFs by targeting or counteracting signals from tumor cells, (2) regulating the activation of CAFs by targeting the CAF population directly, (3) regulating the pro-tumorigenic signals from CAFs, (4) regulating the pro-angiogenic signals from CAFs, and (5) regulating the pro-immune evasion and anti-immune surveillance signals from CAFs (Figure 3). These potential CAF intervention points represent ‘action items’ to ’switch off’ the pro-resistance CAFs within the tumor stroma.

### 4.2. Stromal Normalization and CAF-Targeted Therapy in Combating Resistance to Chemotherapy and Targeted Therapy

CAFs co-operate with tumor cells to drive the progression of the disease [65,66,67]. The progression of the disease can be attributed to this collaboration of CAFs with tumor cells based on several factors and events, either individually or collectively, including (1) the EMT; (2) stemness; (3) response to hypoxia; (4) pro-proliferative and anti-apoptotic signals; (5) immune, metabolic, and ECM reprogramming; (6) metastasis-associated phenotypes; and (7) escape and resistance to therapy.

CAFs have been targeted using both conventional and unconventional modes of disease management in solid tumors, diagnostics, and therapeutics. Although CAFs have been identified using several markers, FAP and alpha-SMA are among the most versatile markers associated with the identification and function of CAFs [87]. FAP has been targeted in tumors for imaging and therapy using several approaches, including immunoconjugates (an antibody–maytansinoid conjugate (mAb FAP5-DM1)), CAR T cells, tumor immunotherapy, vaccines, peptide drug complexes, FAP inhibitors, and antibodies [87,88,89]. The depletion of FAP-positive CAFs enhanced anti-tumor immunity, as reported in several studies [90,91,92,93,94], proving the validity of the target. In fact, co-targeting FAP in combination with tumor-centric FAP-targeting strategies was shown to be more effective [95,96,97]. Anti-FAP antibody sibrotuzumab labeled with 131Iodine has been reported for the treatment of patients with metastasized FAP-positive carcinomas in a phase I dose-escalation study [98]. To test the diagnostic and prognostic value of the imaging of activated fibroblasts, Lindner et al. developed the radiotracers FAPI-01 and FAPI-02 with specific binding to human and murine FAP with a rapid and almost complete internalization [86]. The DOTA-linked compound FAPI-02 with better pharmacokinetic and biochemical properties was tested for quantitative analysis of tracer uptake in 80 patients with 28 different tumor entities (54 primary tumors and 229 metastases). Their study indicated that FAP inhibitors have a promising role as tracers for diagnostic applications in desmoplastic tumors.

CAFs have been targeted using nano or gold particles in a radio-pharmacological manner. CAFs have been reported to be explicitly targeted by nanocarriers with optimized physicochemical properties in liver cancer. Surface-modified nanocarriers with a cyclic peptide binding to the PDGFRβ or mannose-6-phosphate binding to the IGFRII effectively directed the drug to activate CAFs in vivo [99]. Gold nanoparticles measuring 20 nm in diameter inhibited CAF activation by disrupting multicellular communication between the tumor and microenvironment and altering the levels of multiple fibroblast activation or inactivation proteins, such as TGF-β1, PDGF, uPA, and TSP1, secreted by ovarian cancer cells and TME cells [100]. Passive and active strategies for the nanodelivery systems targeting CAFs for improved anti-tumor effect and tumor drug penetration have been summarized elsewhere [101,102]. The recent advancements in targeting CAFs with diagnostic and therapeutic radiopharmaceuticals by applying new radiotheranostic compounds (targeted radionuclide imaging and therapy) using clinically identified biomarkers to improve clinical outcomes are promising [103,104].

CAF targeting has also been studied in rare solid tumors with highly desmoplastic stroma in intrahepatic cholangiocarcinomas [105,106,107]. Mertens et al. reported that navitoclax induced apoptosis in CAFs and in myofibroblastic human hepatic stellate cells but lacked similar effects in quiescent fibroblasts or cholangiocarcinoma cells, arguing for the use of navitoclax (Bcl2/Mcl inhibitor) for destroying CAFs in the TME [108]. In desmoplastic cholangiocarcinoma, the use of light-activated nanohyperthermia has been described to modulate the tumor microenvironment [109]. A recent study employed multifunctional iron oxide nanoflowers decorated with gold nanoparticles (GIONF) as efficient nanoheaters to achieve complete tumor regression following three sessions of mild hyperthermia. CAFs were targeted via preferential uptake of GIONF. A photothermal depletion of CAFs resulted in a significant early reduction in tumor stiffness (normalized tumor stiffness) followed by tumor regression. Katsube et al. employed near-infrared photoimmunotherapy (NIR-PIT) as a novel method of cancer treatment using a highly selective monoclonal antibody (mAb)–photosensitizer conjugate against fibroblast activation protein (FAP)-targeted NIR-PIT, in which IR700 was conjugated to a FAP-specific antibody to target CAFs (CAF-targeted NIR-PIT: CAFs-PIT) [110]. The elimination of CAFs by CAFs-PIT demonstrated that the combination of 5-FU and NIR-PIT caused a 70.9% tumor reduction, while 5-FU alone achieved only a 13.3% reduction, suggesting the recovery of 5-FU sensitivity in CAF-rich esophageal tumors in experimental models.

Yet another classic example of stromal resistance mediated through CAFs is represented by PDAC (pancreatic ductal adenocarcinoma), a disease in which the five-year overall survival for pancreatic cancer is still less than 10%, despite advances in therapeutic modalities [111]. Pancreatic tumors present a highly fibrotic stroma containing activated CAFs, which create an immunosuppressive TME. CAFs secrete immunoregulatory and chemo-attractive factors, preventing tumor-reactive T-cell responses. Gorchs and Kaipe summarized different therapy strategies targeting the CAF–T cell axis, focusing on CAF-derived soluble immunosuppressive factors and chemokines to highlight the strategies that can be used to target CAFs in the context of the capability of heterogeneous CAFs to modulate functions of TILs and myeloid cells in desmoplastic pancreatic ductal adenocarcinomas (PDACs) [111]. Although the CAF-immune cell dialogue is beyond this review’s scope, identifying the immunological functions of different CAF subsets (for example, inflammatory fibroblasts (iCAFs) and myofibroblasts (myCAFs)) that help tumor cells to (1) evade immune surveillance and (2) potentiate immune exhaustion may be essential for the development of an effective combinational treatment for desmoplastic solid tumors.

### 4.3. CAF-Mediated Immune Reprogramming

Since CAFs induce immunotherapy resistance and influence tumor immunity and immunotherapy [112,113], CAFs have been targeted using various modes in anti-cancer immunotherapy [114]. The establishment of mechanisms of CAF-mediated blockade of CD8+ cytotoxic T-cell accumulation in tumors has provided therapeutic opportunities [115]. CAFs crosstalk and co-evolve with cancer stem cells [116]. Therapeutic targeting of the manipulation of cancer stem cells [116,117] and immune-reprogramming using CAFs provides a window of opportunity beyond this review’s scope. In an exceptionally aggressive and treatment-resistant human cancer, the role of dermal fibroblasts in suppressing the tumorigenesis has been documented, which are subsequently converted or activated to CAFs, which are phenotypically and epigenetically different from normal dermal fibroblasts. Flach et al. demonstrated that melanoma cells could stimulate the recruitment of fibroblasts and activate them, resulting in melanoma cell growth by providing both structural (extracellular matrix proteins) and chemical support (growth factors). Thus, CAFs collaborate with melanoma cells and resist drug therapy [118]. Kinugasa et al. demonstrated that established CAFs enhance tumor growth in vivo in B16 melanoma-bearing mice. These CAFs strongly express CD44 in the hypovascular and hypoxic areas of the TME or following treatment with angiogenesis inhibitors. CD44 expression in CAFs maintains the stemness of cancer stem and initiating cells via direct interaction and is involved in drug resistance [119]. Bellei et al. reviewed the melanoma–CAF dialogue based on TGF-beta, MAPK, Wnt/beta-catenin, and Hippo signaling [120]. It makes sense that the activation of the Wnt/beta-catenin pathway may lead to the expression of CD44 (target gene) in CAFs and signaling for the stemness-driven drug resistance of the disease.

### 4.4. CAF-Mediated EMT and ECM Reprogramming

The induction of stemness and EMT are two main phenotypic steps of the multi-step process of metastasis in solid tumors. It is worth mentioning that stemness and morphological transition between the epithelioid and fibroblastoid features of tumor cells are closely integrated, especially in the types of solid tumors, wherein cancer cells with fibroblastoid morphological changes exhibit increased motility and invasiveness due to decreased cell–cell adhesion, reminiscent of EMT in many solid tumors. In promoting metastasis, the silencing of DNMT1 is correlated with the enhancement of the induction of EMT and the CSC (cancer stem cells) phenotype in prostate cancer cells [121]. functional connection of CAFs in EMT via DNA methylation was presented in the study by Pistore et al. in advanced prostate cancer. The secreted factors in conditioned media from CAFs explanted from two unrelated patients were found to stimulate concurrent DNA hypo- and hypermethylation required for EMT and stemness in PC3 and DU145, indicating that CAF-released factors induce genome methylation changes required for EMT and stemness in EMT-prone cancer cells [122]. One such secreted factor from CAFs was reported to be TGF-beta in several solid tumors [123,124,125,126]. Cardenas et al. demonstrated that TGF-beta stimulated EMT and that metastasis catalyzed the global DNA hypermethylation changes in the epithelial ovarian cancer cells, while the DNMT inhibitor blocked the hypermethylation and EMT [127]. In fact, TGF blockade has been reported to improve the distribution and efficacy of therapeutics in breast carcinoma by normalizing the tumor stroma interstitial matrix by decreasing collagen I content to improve the intratumoral penetration of both a low-molecular-weight conventional chemotherapeutic drug and a nanotherapeutic Doxil [128]. CAF-induced epigenetic modification of cancer cells leading to drug resistance could be a potential way to design a CAF-targeted inclusive strategy for therapy in the future. In line with the association of CD44 expression in CAFs as discussed above, CAFs have been shown to secrete soluble factors belonging to Wnt family members and the Wnt/beta-catenin pathway. WNT16B and SFRP2 activated the canonical Wnt pathway in tumor cells and induced cytotoxic chemotherapy resistance in prostate cancer [129,130]. In colorectal cancers, chemoresistance in cancer-initiating cells was also increased by CAFs. Lotti et al. conducted a comparative analysis of matched colorectal cancer specimens from patients before and after cytotoxic treatment to demonstrate a significant increase in CAFs. Chemotherapy-treated human CAFs promoted cancer-initiating cell self-renewal via IL-17A, and IL-17A was found to be overexpressed in colorectal CAFs in response to chemotherapy, as validated directly in patient-derived specimens without culture [131]. The study directly proved that CAFs respond to therapy in favor of tumor cells and strongly supported the unmet need to include a CAF-directed therapy towards the ‘normalization’ of the ‘resistant stroma’.

### 4.5. CAF-Mediated Metabolic Reprogramming and Hypoxic Response

As cancer cells biochemically reprogram their metabolism as their hallmark, they generate lactic acid from glucose or glutamine. Cancer cells export lactic acid out, preventing intracellular acidification causing increased lactate levels and an acidic pH level in the extracellular milieu [71]. Lisanti et al. reviewed metabolic coupling between mitochondria in cancer cells and catabolism in stromal fibroblasts [132]. Unlike tumor cells, CAFs are catabolic by default. CAFs donate L-lactate, ketones, glutamine, other amino acids, and fatty acids to cancer cells to metabolize via their TCA cycle and oxidative phosphorylation. This metabolic coupling explains how metabolic energy and biomass are supplied by the CAFs to cancer cells. Lisanti et al. demonstrated that catabolic metabolism and the glycolytic reprogramming in the CAFs (a loss of caveolin-1 and an increase in MCT4 in CAF) are influenced by oncogenes in epithelial cancer cells, including BRCA1-deficient breast and ovarian cancer cells, in concert with the TME [133]. Interestingly, both oncogenic activation (of RAS, NFkB, and TGF-β) and loss of the tumor suppressor (BRCA1) have comparable effects on CAF. Arguably, such a ‘metabolic symbiosis’ could provide an explanation for the ‘fibroblast addiction’ or ’metabolic parasites’ in primary and metastatic tumor cells [134] and could present a target for therapy, wherein CAFs could be decoupled from tumor cells. The ensuing hypoxic environment adds yet another layer to the chemoresistance [135] due to the influence of low pH on the cytotoxicity of paclitaxel, mitoxantrone, and topotecan [136]. Hypoxia is a fact of life for cancer cells in solid tumors [137,138,139]. As a critical player in the development of drug resistance, it is most logical that CAFs will have a direct role in modulating drug sensitivity or action in a hypoxic environment. CAFs secrete elements of different angiogenic and immunogenic signaling pathways, including VEGF and T-cell-mediated cytotoxicity, respectively, under hypoxic conditions [140,141,142]. Masamune et al. reported hypoxia-induced pro-fibrogenic and pro-angiogenic responses in pancreatic stellate cells [143]. Pancreatic stellate cells expressed several angiogenic molecules, including VEGF receptors, angiopoietin-1, and Tie-2. Studying the effects of hypoxia and conditioned media of hypoxia-treated pancreatic stellate cells on cell functions and on human umbilical vein endothelial cells, Masamune et al. demonstrated that hypoxia accelerated migration, type I collagen expression, and VEGF production in pancreatic stellate cells. Conditioned media of hypoxia-treated pancreatic stellate cells induced migration of pancreatic stellate cells, which was inhibited by the anti-VEGF antibody. Conditioned media of hypoxia-treated pancreatic stellate cells, on the other hand, induced endothelial cell proliferation, migration, and angiogenesis in vitro and in vivo. In line with the above study, endothelial cells co-cultured with CAFs under hypoxia or exposed to the conditioned medium of hypoxic CAFs have been shown to sprout significantly more than the normoxic counterpart in breast cancers [144]. These data functionally connect CAF activity with the tumor angiogenesis and resistance or metastasis progression associated with tumor cell phenotypes under hypoxic conditions, strengthening the argument in favor of a CAF-inclusive treatment strategy.

### 4.6. CAF-Based NIH Clinical Trials

The clinical trials targeting CAFs in solid tumors are based on antagonizing CAF functions. Overall, trials can be divided into (1) reprogramming of CAFs, (2) inhibition of CAF functions, (3) targeting of CAF-mediated desmoplasia, and (4) CAF-specific immunotherapy. The details of these trials are presented elsewhere [145]. FAP proteins are some one of the common targets in the clinical trials related to CAFs. Accordingly, anti-FAP vaccination has been reported in various tumor models [146]. The other aspects of CAF-related trials involve targeting the interactions between tumor-promoting CAFs and the surrounding microenvironment and reprogramming CAFs into quiescent fibroblasts or reprogramming tumor-promoting CAFs into tumor-restraining CAFs, as presented in detail elsewhere [147].

Reviewing the CAF-associated clinical trials on the ClinicalTrials.gov site, we present 17 trials involving CAFs (Table 1). These studies have used various aspects of markers or functions of CAFs, the culture or co-culture of CAFs, testing of drug combinations targeting CAFs, or disease detection using CAF-based radiochemicals, as mentioned before. The studies ranged from observational to interventional or treatment to open-label. The primary purposes also varied from diagnostic to treatment to exploratory basic science. Most of the trials were conducted in disease conditions of advanced or malignant neoplasms of solid tumors in adults. Table 1 presents the relevant ongoing and completed trials involving CAFs posted in ClinicalTrials.gov (as of February 2022). The studies were performed in advanced or malignant neoplasms of solid tumors in adults, including hepatocellular, lung, breast, and pancreatic cancers. The observational study, *NCT01549275,* is among the two completed studies. This ‘case-only’ prospective study enrolled 105 patients with hepatocellular carcinomas. The prospective study evaluated the success rate of the primary culture of hepatocellular carcinoma cells and CAFs from the residual specimens in routine fine-needle aspiration of hepatic tumors and the potential application of this method as an additional tool for personalized treatment of patients. The primary outcome measure was to find the correlation between the growth speeds of the cultured cells and the AJCC TNM stage (7th Eds) at entering the study within a time frame of 28 days after plating of cells. The other completed study, *NCT02161523*, tested the impact of lung CAFs on mast cell activation in lung cancers. This prospective observational study involved fewer patients than the first, with non-small cell lung carcinomas. This study evaluated the paracrine function of CAFs and directly measured the factors in the lung TME (which includes other cells such as fibroblasts that are attributed to mast cell activation). The trial was conducted to determine whether CAF cells derived from lung tumors, together with the lung cancer cells or microvesicles derived from these cells, are able to stimulate mast cells to degranulate or release various cytokines and chemokines. CAFs were co-cultured with both lung cancer cell lines (A-549) or microvesicles derived from these cells and the human mast cell line (LAD2). The collected supernatants were used to determine degranulation and cytokine release from these mast cells as the primary outcome by measuring the levels of b-hexosaminidase (a marker for mast cells degranulation) and the cytokines levels within a time frame of 1–2 weeks.

In addition to the studies covering the functions of CAFs, studies have also been undertaken to utilize CAFs in developing resistance to chemotherapy in solid tumors in combination with tumor-centric therapy. The role of CAFs in the reprogramming of the ECM by altering the state of hyaluronic acid and the consequences for the tumor ECM and tumor vasculature have been presented in several reviews [148,149,150]. Hyaluronan synthase 2 has been reported to be expressed in CAFs to promote invasion in oral cancers [151]. An in vitro evaluation of simultaneous targeting of tumor cells and CAFs with a paclitaxel–hyaluronan bioconjugate was carried out in non-melanoma skin cancers by Bellei et al. [152].

CAFs can provide a physical and vascular barrier, depriving the tumor of TILs and protecting against chemotherapy. Hence, the ‘normalization of the TME’ has been proposed as a viable target of treatment, especially in solid tumors with high desmoplastic reactions such as chemotherapy-resistant advanced PDAC. The *NCT03481920* study was a pilot trial of PEGPH20 (pegylated hyaluronidase) in combination with avelumab (anti-PD-L1 MSB0010718C) in chemotherapy-resistant pancreatic cancers. The purpose of this multi-center, open-label, non-randomized early phase 1 trial (intervention–treatment) was to evaluate the pharmacodynamics, safety, and efficacy of PEGPH20 in combination with avelumab in adult patients with chemotherapy-resistant advanced or locally advanced PDAC. The study tested the hypothesis that elimination of HA (hyaluronic acid) in the pancreatic TME mediated by PEG PH20 would result in increased tumor vascularization and vessel patency, as well as stromal remodeling with increased immune infiltration. The activity of immune checkpoint inhibitors such as avelumab was facilitated by at least two mechanisms, including an increase in drug delivery and increasing immune infiltration.

Another function of CAFs within the TME is associated with EMT and mesothelial–mesenchymal transition in the context of peritoneal dissemination. Peritoneal dissemination is a frequent metastatic route for cancers of the ovary and gastrointestinal tract. Solid tumors in the abdomen, such as gastric, colorectal, and ovarian cancers, commonly disseminate via a transcoelomic route, an event associated with a poor prognosis [153]. Metastases are influenced by CAFs, a cell population that derives from different sources. CAFs are known to derive from mesothelial cells via mesothelial–mesenchymal transition during a peritoneal metastasis [154]. A type II EMT, known as mesothelial–mesenchymal transition (MMT), occurs after peritoneal damage. Myofibroblast conversion of mesothelial cells contributes to peritoneal fibrosis associated with peritoneal dialysis and post-surgical peritoneal adhesion. In a recent report, Gordillo et al. reported that MMT contributes to the generation of CAFs in locally advanced primary colorectal carcinomas [155]. In a prospective recruiting study, *NCT03777943*, the role of the peritoneal microenvironment in the pathogenesis and spread of colorectal carcinomatosis (MMT) was evaluated. The study investigated the extent and role of MMT and CAFs in the pathogenesis of colorectal peritoneal carcinomatosis. The primary outcome of the study was the analysis and sampling of peritoneal tissue via immunohistochemistry of CD44, integrins, ICAM-1, hyaluronate, and VCAM-1 (adhesion molecules); calretinin, mesothelin, WT1, cytokeratins, and E-cadherin (mesothelial markers); α-SMA, FAP, and podoplanin (CAF specific markers); and PDGF, VEGF, and other angiogenesis-related markers within 6 months after the collection of the samples from patients presenting with colorectal peritoneal carcinomatosis.

Normal residential fibroblasts become activated by tumor cells and are sources of CAFs. Fibroblast activation protein (FAP) is one of the emerging reliable markers of CAFs [1,156,157]. FAP is a transmembrane protein expressed on CAFs and has been shown to be differentially present on a number of solid tumors as a marker of CAFs. FAPs have been exploited for certain diagnostic and treatment purposes in clinical trials. Radionuclide-labeled fibroblast activation protein inhibitors (FAPI) targeting FAP as a tracer for PET imaging have been tested for targeted diagnosis and treatment of cancer. Although the function of FAP is yet to be established, imaging studies have shown that FAP could be detected with FAPI PET/CT.

The interventional open-label clinical Trial, *NCT04554719,* with a primary diagnostic purpose, studied the clinical application of FAP PET/MRI for diagnosis and staging. This prospective trial, in which 100 patients with malignant neoplasm participated, was based on the background that FAP is overexpressed in CAFs, which is closely related to tumor growth, invasion, metastasis, immunosuppression, and prognosis, while the expression level of FAP in normal tissues and organs is very low. The trial used integrated PET/MR and PET/CT with the agent 68Ga-FAPI ((gallium-68 (68Ga)–FAPI) as a new novel positron tracer and the conventional imaging agent F-18 (fluorodeoxyglucose 18F-FDG) to diagnose and stage various cancers with the aim of making up for the deficiency in FDG–PET imaging in the diagnosis and staging of certain cancers. Another clinical trial was named *NCT04621435*, with a primary purpose of diagnosis based on FAP-2286, a peptidomimetic molecule that binds to FAP. The study was a single-arm prospective trial that evaluated the ability of a novel imaging agent gallium-68-labeled (68Ga-) FAP-2286 (68Ga-FAP-2286) to detect metastatic cancer in adults with solid tumors using 68Ga-FAP-2286 tracer. In contrast to the above studies, the phase 1/2 recruiting trial *NCT04939610* (LuMIERE) tested 68Ga-FAP-2286 and 177Lu-FAP-2286 for the primary purpose of treatment. This multicenter, open-label, non-randomized study investigated the safety, tolerability, pharmacokinetics, dosimetry, and preliminary activity of 177Lu-FAP-2286 in 170 participating patients with advanced solid tumors. Phase 1 of the study evaluated the safety and tolerability of 177Lu-FAP-2286 and determined the recommended phase 2 dose (RP2D) in patients with advanced solid tumors. Phase 2 of the study evaluated the objective response rate (ORR) in patients with specific solid tumors. *NCT04459273*, a prospective exploratory trial, studied the PET biodistribution of 68Ga-FAPI-46 (FAPi PET/CT) in patients with a wide range of solid tumors. The study investigated how an imaging technique called 68Ga-FAPi-46 PET/CT can determine where and to which degree the FAPi (fibroblast activation protein inhibitor) tracer (68Ga-FAPi-46) accumulated in normal and cancer tissues in patients. The trial sought to define the biodistribution of gallium Ga 68 fibroblast activation protein inhibitor (FAPi)-46 (68Ga-FAPi-46) in normal and cancer tissues of patients with various malignancies.

CAFs are responsible for metabolic reprogramming in the TME, involving ROS in certain solid tumors [123,158]. The *NCT01878695* trial, sponsored by the Sidney Kimmel Cancer Center at Thomas Jefferson University, on the contrary, plans to assess the feasibility of evaluating the effects of n-acetylcysteine on tumor cell metabolism by determining the changes in expression of caveolin -1 and MCT4 in CAFs in pre- and post-therapy breast tissue samples treated with NAC (N-acetyl derivative of the naturally occurring amino acid, L-cysteine). This interventional open-label clinical trial with a primary purpose of treatment is a pilot study of anti-oxidant supplementation with N-acetyl cysteine in stage 0/I breast cancers.

From the detailed overview of the clinical trials presented above, it is apparent that CAFs offer a reasonable target that is complementary to the tumor-centric management of the disease. CAFs and their markers offer a basic scientific, diagnostic, and complementary (companion) treatment opportunity, more so in advanced solid tumors of breast, pancreas, peritoneal, and lung carcinomatosis. One remarkable fact emerging from the current literature is the conspicuous lack of basic and translational data regarding gynecological malignancies such as ovarian and endometrial cancers. The role of the endometrial stroma in pathogenesis is known, and human endometrial stromal cells have been found to express CD90, CD10, and CD140b [159]. In situ staining of the human myometrium and endometrium demonstrated heterogeneous staining for Thy 1. Freshly derived fibroblast strains from the myometrium and endometrium showed heterogeneous Thy 1 expression [160]. In fact, the prognostic significance of the tumor/stromal ratio (TSR), which is established in several solid tumors, has also been reported in endometrial carcinoma [161]. In their first attempt to characterize the fibromyxoid stromal reaction (desmoplasia) and a lymphocytic infiltrate, Espinosa et al. sought to find out the relationship between the desmoid-type fibromatosis stromal signature and the presence of desmoplasia [162]. Although a study by Micke et al. failed to find a significant difference in the Kaplan–Meier plots of the overall survival between stroma-rich and stroma-poor groups of endometrial patients [163], Espinosa et al. demonstrated that desmoplasia correlated positively with the desmoid-type fibromatosis expression signature, and stromal signatures have significant clinicopathological associations. Considering the (1) presence of fibroblasts in the uterine stroma, (2) the role of CAFs in the neoplastic transformation and progression of the disease, and (3) the significance of the stromal signature in endometrial cancers, the inadequacy of data and lack of trials in endometrial cancers remain puzzling. The conspicuous lack of information on the role of CAFs in the development of drug resistance in endometrial tumors can be explained by (1) the absence of relevant data regarding the characterization of CAFs based on a drug resistance condition in the context of different pathological parameters, genomic alterations, and outcome data and (2) the absence of a correct model system. It is understood that a bulk of endometrial cancers are detected early, whereby patients undergo surgical resection. Drug resistance conditions in the advanced or late stages in endometrial cancers are rarely presented where the tumor tissue can be accessed surgically. The characterization of endometrial CAFs and their presentation in the context of pathological parameters, genomic alterations, and outcome data in the future will pave the pathway for developing a model to test the functions of endometrial CAFs in a drug resistance scenario. However, it should be emphasized that we are only beginning to understand the complexity of the functions of CAFs and their function-specific markers in solid tumors. The future will unveil the clinical utility of the knowledge.

## 5. Forward Thinking

Resistance to therapy is a pro-tumorigenic event. CAFs are employed by tumor cells to create a pro-tumorigenic microenvironment following treatment. Resistance to treatment is the outcome of a highly efficient adaptive strategy orchestrated by cancer cells via reprogramming of their default signals, which co-occurs with the reprogramming of every component of the TME in their favor, including CAFs. Such an opportunistic event allows the tumor cells to gain contextual survival and progressive metastatic advantages. Tumor–stroma co-evolution can lead to the development of drug resistance. The liaison between CAFs and tumor cells can be viewed as a *bête noire* of therapy. Thus, CAFs as the critical or indispensable components of stromal resistance to treatment are the most logical targets within a tumor that has eventually progressed despite therapy. As the roles of CAFs in several aspects of tumor progression and the development of drug resistance are unfolding, the notion of CAFs being friend or foe [164] is evolving. CAFs are neither heroes nor villains [165]. CAFs are less cause for panic but demand more urgent action, especially in scenarios involving a therapy-resistant progressing tumor. We need to know more about how CAFs form multi-faceted support systems for drug-resisting progressing tumors to exercise that knowledge in empowering the management of the disease by including CAF-directed stromal-targeting agents [166] in the arsenal of targeted therapy options.

The roles of CAFs in several common and rare tumors, as presented above, give us an idea about their role in (1) tumor progression and (2) modes of development of resistance to treatment. It has to be recognized that the heterogeneity of CAFs could be associated with better outcomes or response to therapy as opposed to their pro-tumor actions. Bhattacharjee et al. demonstrated direct CAF–tumor interactions as a tumor-promoting mechanism, mediated by myCAF-secreted hyaluronan and inflammatory-iCAF-secreted HGF [167]. The pro-tumorigenic effects seen in their study were opposed by myCAF-expressed type I collagen, which suppressed tumor growth by mechanically restraining the tumor spread. Their study directly indicated that there is a scope for the therapeutic maneuvering of CAF function in favor of the patient outcome by targeting specific signals for the tumor-promoting function of CAFs, while promoting the myCAF-expressed type I collagen. This report, similar to other articles [25], indicated the possibility of establishing therapeutically targetable CAF-subtype-specific mediators for future treatment directed towards stromal normalization of desmoplastic tumors.

The study of CAFs and their origins, markers, and functions in the development of drug resistance can be conducted in tumors of the pancreas, breasts, stomach, esophagus, colorectal, prostate, and lungs, as well as melanoma, head and neck squamous cell carcinoma, renal cell carcinoma, and cholangiocarcinomas. Understandably, CAF-inclusive clinical trials are instituted in these organ cancers via various modes of intervention [145]. In a recent review, Koustoulidou et al. presented an overview of several modes of intervention using (a) anti-FAP mAbs (b)-engineered T-cells expressing an FAP-recognizing mAb (e.g., CAR-T cells) to target FAP+ CAFs, which resulted in their immune-cell-mediated destruction and removal, (c) enzymatic breakdown of hyaluronic acid to remodel the ECM for better accessibility of drugs to immune cells with tumor parenchyma, (d) blocking of CAF activity by interleukin-6, (e) transformation of CAFs into the quiescent state by vitamin D, and (f) blocking of CAF-induced metabolic reprogramming of tumor cells [104]. Recently, organ-specific subtypes of CAFs have been identified and associated with different functions in aiding and abetting tumor cells, as reported by Engelman’s group [82]. Their study will encourage others to study the organ-specific roles of subtypes of CAFs and their particular modes of action in the progression of tumors.

As we evaluate the participatory role of CAFs in the development of drug resistance in solid tumors, we will have to design a workable model to test our hypotheses; ideally on a patient-to-patient basis in the context of each patient’s unique genomic alteration(s). The possibility of co-targeting CAFs and testing whether they will have a clinical benefit towards managing the burden of resistance in the future will rely on such a model system, which will accommodate tailored testing of the roles of patient-derived CAFs in the context of both tumors cells and other components of the TME. Although much effort is still needed to translate CAF-directed anti-cancer strategies from the bench to the clinic, the future will establish the specific modes of action of CAFs in particular organ-type solid tumors, paving the way for CAF-inclusive personalized therapy in solid tumors.

In summary, we will require actionable insights into the functions of CAF subtype(s) to incorporate CAF-directed therapy in clinics. Actionable information on the CAF subtypes in the context of their functions will be needed regarding (1) specific clusters associated with immunosuppression and immunotherapy resistance [26], (2) therapeutically targetable CAF-subtype-specific mediators [25], (3) the Hedgehog pathway inhibition by a smoothened antagonist, LDE225-mediated differential activation of myCAFs or iCAFs leading to alterations of cytotoxic T cells and Tregs [30], (4) IL1B blocking agents to counteract the iCAF-mediated pro-tumorigenic actions associated with tolerance to cytotoxic drugs [33], and (5) differential targeting of tumor-promoting CAF mediators while preserving the specific anti-tumor functions, for example in the way type I collagen may ‘normalize’ stroma from tumor-promoting to tumor-restricting phenotypes [167].

## 6. Take-Home Message

The undeniable subpopulation-specific functions of CAFs in tumor growth, progression, and drug or immunotherapy resistance directly provide evidence for the therapeutically targetable role of CAFs. The aim of normalization of the TME by targeting CAFs remains unmet. CAFs are heterogeneous and organ-type-specific in origin, markers, and function. Hence, the best way to develop a ‘workable hypothesis’ for the functions of CAFs would be to generate strictly organ-specific experimental evidence. It is imperative to know the functions of specific signals from different CAF subtypes within the TME of organ-type cancer(s). We can exploit the information for (1) targeting of the pro-normalization signals from CAFs while attenuating the pro-growth progression and immunosuppressive CAF signals and (2) identifying potential CAF markers to investigate the mechanisms underlying the roles of CAFs in the TME.

## 7. Conclusions

In the era of precision medicine, which offers clinicians to treat patients with genomics-guided matched drug combination(s), the cure still remains an exception and not the rule. CAF-mediated development of resistance is the bête noire of chemotherapy and targeted therapy as CAF directly supports the development of resistance. The state-of-art management of today’s disease does not necessarily include a CAF-inclusive therapy. We are just beginning to appreciate that the knowledge about the CAF functions and inhibition is critical in managing the disease towards developing a CAF-inclusive therapy.

## Figures and Tables

**Figure 1 cancers-14-01519-f001:**
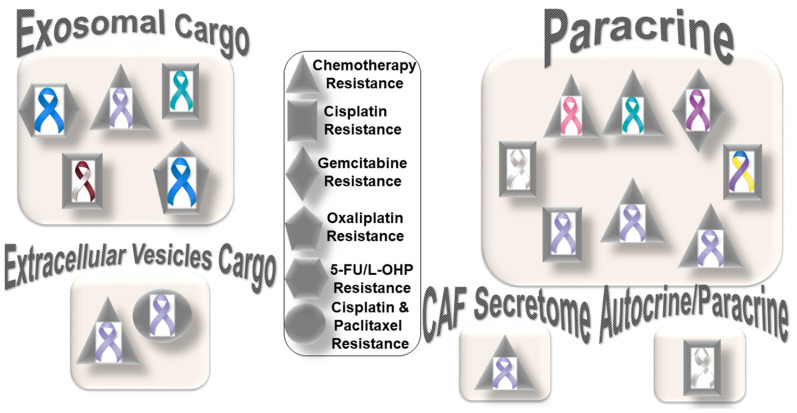
Distribution pattern of types of resistance to chemotherapy based on specific mediators of CAF functions in solid tumors: The four mediators employed by CAFs to orchestrate the development of resistance to chemotherapy are presented in the cartoon. The most common mode of interaction is paracrine, wherein CAFs signal to either tumor cells or other components of the TME via characteristic secretome. In addition to the involvement of the characteristic secretome, exosomal cargos delivering different miRNAs that target various cell signaling proteins are common mediators of CAF actions. Among different organ cancers, gastric cancers have been reported to be the most common tumors in which CAFs are involved in the development of resistance to chemotherapy. The sizes of the boxes indicate the number of studies in each box. The shapes indicate the types of resistance in different tumors (inset). L-OHP is a new derivative of oxaliplatin; 5-FU is fluorouracil. Organ tumors are indicated by their respective ribbon colors. Head and neck cancer: white and burgundy; stomach cancer: periwinkle blue; colon cancer: dark blue; ovarian cancer: teal; lung cancer: white or pearl; breast cancer: pink; pancreatic cancer: purple; bladder cancer: blue, yellow, and purple.

**Figure 2 cancers-14-01519-f002:**
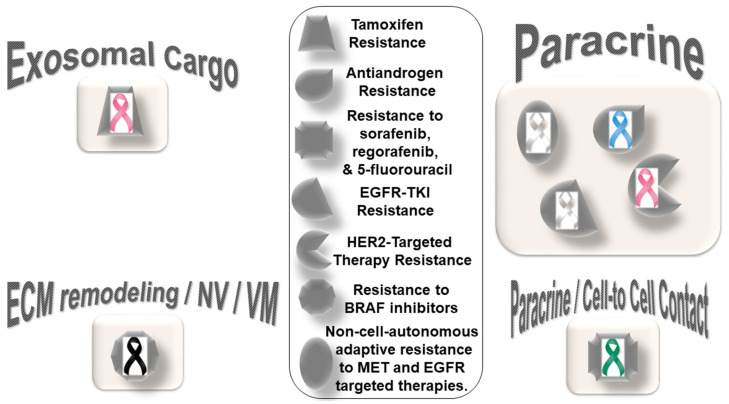
Distribution pattern of types of resistance to targeted therapy based on specific mediators of CAF functions in solid tumors: The four types of mediators of action employed by CAFs to orchestrate the development of resistance to targeted therapy are presented in the cartoon. The most common mode of interaction is paracrine, wherein CAFs signal to either tumor cells or other components of the TME via characteristic secretome. In addition to the involvement of characteristic secretome, exosomal cargos delivering different miRNAs that target various cell signaling proteins are common mediators of CAF action. The sizes of the boxes indicate the number of studies in each box. The shapes indicate the types of resistance in different tumors (inset). Organ tumors are indicated by their respective ribbon colors. Lung cancer: white or pearl; skin cancer: black. liver cancer: emerald green; breast cancer: pink; prostate cancer: light blue.

**Figure 3 cancers-14-01519-f003:**
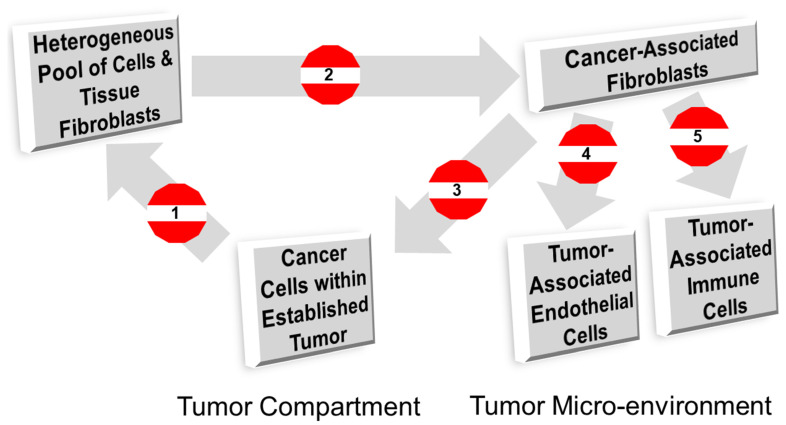
Strategic opportunities to regulate CAF functions in an established or progressing solid tumor. The strategic points to control the function of CAFs are (1) prevention of activation of CAFs by targeting or counteracting signals from tumor cells, (2) regulating the activation of CAFs by targeting the CAF population directly, (3) regulating the pro-tumorigenic signals from CAFs, (4) regulating the pro-angiogenic signals from CAFs, and (5) regulating the pro-immune evasion and anti-immune surveillance signals from CAFs. These strategic points represent ‘action items’ to ‘switch off’ the pro-resistance CAFs within the tumor stroma.

**Table 1 cancers-14-01519-t001:** Trials involving CAFs in cancers as posted in ClinicalTrials.gov (Updated on February 2022) are represented. The 17 trials include various organ cancers in solid tumors, including breast, colorectal, prostate, lung, pancreatic, hepatocellular, ovarian, and oral carcinomas.

Clinical Trials.gov Identifier and Sponsor	Title	Recruitment Status: Study Start Date and Study Completion Date	Condition or Disease	Study Design			
				Study Type	Enrollment	Observational/Intervention Model:	Time Perspective and Primary Purpose
***NCT*** ***01549275***; Kaohsiung Medical University Chung-Ho Memorial Hospital, Taiwan	Primary Cell Culture of Hepatic Tumorous Cells From Routine Fine-Needle Aspiration	Completed: Study Start Date; April 2010; Study Completion Date: July 2013	Hepatocellular Carcinoma	Observational	105 participants	Case-Only	Prospective
***NCT*** ***02161523***; Meir Medical Center, Kfar Saba, Israel	The Impact of Lung Cancer-Derived Fibroblasts on Mast Cell Activation	Completed: Study Start Date; 1 July 2014; Study Completion Date: 1 June 2015	Lung Cancer	Observational	20 participants	Other	Prospective
***NCT*** ***03481920***; PH Research, S.L. Madrid, Spain	A Pilot Trial of PEGPH20 (Pegylated Hyaluronidase) in Combination With Avelumab (Anti-PD-L1 MSB 0010718C) in Chemotherapy Resistant Pancreatic Cancer	Terminated: Study Start Date; 10 January 2018; Study Completion Date: 10 June 2019	Pancreatic Ductal AdenocarcinomaPancreatic Cancer; Drug: PEGylated Recombinant Human Hyaluronidase (PEGPH20) Drug: Avelumab; Early Phase 1	Intervention /treatment (Open Label)	7 participants	Single Group Assignment	Not Mentioned
***NCT*** ***03777943***; University Ghent, GIHeelkunde, University Hospital, Ghent, Belgium	Role of the Peritoneal Microenvironment in the Pathogenesis and Spread of Colorectal Carcinomatosis (MMT)	Recruiting: Study Start Date; 1 November 2017; Estimated Study Completion Date: December 2020	Peritoneal Carcinomatosis	Observational: Intervention/treatment; Procedure: Sampling peritoneal tissue	50 participants	Other	Prospective
***NCT*** ***04554719***; Wuhan Union Hospital, China	Clinical Application of Fibroblast Activation Protein PET/MRI for Diagnosis and Staging in Malignant Tumors	Recruiting: Study Start Date; 22 May 2020; Estimated Study Completion Date: 21 December 2022	Malignant Neoplasm	Interventional (Clinical Trial)	100 participants	Single Group Assignment: Intervention /treatment Drug: 68Ga-DOTA-FAPI Device: PET/MR Device: PET/CT	Primary Purpose: Diagnostic
***NCT*** ***04939610***; Clovis Oncology, Inc. USA	LuMIERE: A Phase 1/2, Multicenter, Open-Label, Non-Randomized Study to Investigate Safety and Tolerability, Pharmacokinetics, Dosimetry, and Preliminary Activity of 177Lu-FAP-2286 in Patients With an Advanced Solid Tumor: A Study of 177Lu-FAP-2286 in Advanced Solid Tumors (LuMIERE)	Recruiting: Study Start Date; 14 June 2021; Estimated Study Completion Date: 1 June 2026	Solid Tumor	Intervention/treatment; Interventional (Clinical Trial); Drug: 68Ga-FAP-2286 Drug: 177Lu-FAP-2286; Phase 1 Phase 2	170 participants	Sequential Assignment /Non-Randomized	Primary Purpose: Treatment
***NCT*** ***04621435***; Thomas Hope, University of California, San Francisco, Clovis Oncology, Inc. USA	Imaging of Solid Tumors Using 68Ga-FAP-2286	Recruiting: Study Start Date; 14 December 2020; Estimated Study Completion Date: 31 December 2023	Solid Tumors, Adult Metastatic Cancer	Intervention/treatment: Interventional (Clinical Trial) Drug: Gallium-68 labelled (68Ga-) FAP-2286Procedure: Positron Emission Tomography (PET) imaging: Phase 1	65 participants	Parallel Assignment; Allocation: Non-Randomized	Primary Purpose: Diagnostic
***NCT*** ***04459273***; UCLA Jonsson Comprehensive Cancer Center, Cancer Center at the University of California Los Angeles, USA	Prospective Exploratory Study of FAPi PET/CT With Histopathology Validation in Patients With Various Cancers (FAPI PET RDRC): PET Biodistribution Study of 68Ga-FAPI-46 in Patients With Different Malignancies: An Exploratory Biodistribution Study With Histopathology Validation	Recruiting: Study Start Date; 27 August 2020; Estimated Study Completion Date: 1 July 2024	Bladder Carcinoma, Cervical Carcinoma, Cholangiocarcinoma, Hematopoietic and Lymphoid Cell Neoplasm, Hepatocellular Carcinoma, Malignant Adrenal Gland Neoplasm, Malignant Brain Neoplasm, Malignant Pleural Neoplasm, Malignant Skin Neoplasm, Malignant Solid Neoplasm, Malignant Testicular Neoplasm, Malignant Thymus Neoplasm, Neuroendocrine Neoplasm, Thyroid Gland Carcinoma, Urothelial Carcinoma	Interventional (Clinical Trial); Procedure: Computed Tomography Drug: Gallium Ga 68 FAPi-46 Procedure: Positron Emission Tomography;	30 participants	Intervention Model: Single Group Assignment; (Open Label)	Prospective Exploratory Study Primary Purpose: Basic Science
***NCT*** ***01878695***; Sidney Kimmel Cancer Center at Thomas Jefferson University, USA	Pilot Study of Anti-Oxidant Supplementation With N-Acetyl Cysteine in Stage 0/I Breast Cancer (NAC)	Completed: Actual Study Start Date: 26 July 2012; Actual Study Completion Date: 14 May 2015	Stage 0/1 Breast CancerPost BiopsyPre-surgery; Drug: IV/oral N-acetyl-cysteine; Phase 1	Interventional (Clinical Trial)	13 participants	Single Group Assignment; (Open Label)	Primary Purpose: Treatment
***NCT*** ***05196334***; Herlev Hospital, Herlev, Copenhagen, Denmark	Pharmacotyping of Patient-derived Pancreatic Cancer Organoids From Endoscopic Ultrasound-Guided Biopsy as a Tool for Predicting Oncological Response	1 July 2021 and 31 December 2024	Pancreatic Cancer	Observational	40 participants	Cohort	Prospective
***NCT*** ***05034146***; Zhongnan Hospital of Wuhan University, Wuhan, Hubei, China	The Diagnostic Efficiency of 68Ga-FAPI PET/CT in Malignant Tumors	23 February 2021 and 28 February 2023	Fibroblast Activation Protein Inhibitor PET/CT Malignant Neoplasm	Interventional (Clinical Trial);	100 participants	Intervention Model: Single Group Assignment	Primary Purpose: Diagnostic
***NCT*** ***04504110***; Peking Union Medical College Hospital; Beijing, China	A Prospective Study to Evaluate 68Ga-FAPI-04 and 18F-FDG PET/CT in Patients With Epithelial Ovarian Cancer: Compared With Histological Findings	5 August 2020 and August 2021	Epithelial Ovarian Cancer	Interventional (Clinical Trial); Phase 2	30 participants	Intervention Model: Single Group Assignment	Diagnostic
***NCT*** ***05030597***: Zhongnan Hospital of Wuhan University, Wuhan, Hubei, China	Exploring the Application Value of PET Molecular Imaging Targeting FAP in Oral Squamous Cell Carcinoma	15 September 2021 and 31 December 2023	PET/CT FAPI Oral Cancer	Interventional (Clinical Trial)	100 participants	Intervention Model: Single Group Assignment	Diagnostic
***NCT*** ***05209750***; The Netherlands Cancer Institute	Pilot Study of FAPI PET/CT for Locoregional (Re)Staging of Lymph Nodes in Colorectal Carcinoma	February 2022 and August 2024	Colorectal Cancer	Interventional (Clinical Trial)	30 participants	Intervention Model: Single Group Assignment	Diagnostic
***NCT*** ***02587793***; Kaohsiung Medical University Chung-Ho Memorial Hospital; Taiwan	Primary Culture of Residual Specimens Obtained From Aspiration of Hepatic Tumor to Predict the Prognosis of the Patients	October 2014 and 31 July 2018	Hepatocellular Carcinoma	Observational	208 participants	Observational Model: Case-Only	Prospective
***NCT*** ***05064618***; Nagoya University Hospital, Nagoya, Aich, Japan	Phase I/II Investigator-initiated Clinical Trial of MIKE-1 With Gemcitabine and Nab-Paclitaxel Combination Therapy for Unresectable Pancreatic Cancer	23 August 2021 and 30 April 2025	Pancreatic Cancer	Interventional (Clinical Trial) Phase 1 Phase 2	55 participants	Intervention Model: Sequential Assignment; Non-Randomized	Treatment
***NCT*** ***02307058***; University of Miami, Florida, USA	A Phase II Randomized Trial of MRI-Guided Prostate Boosts Via Initial Lattice Stereotactic vs. Daily Moderately Hypofractionated Radiotherapy—The Miami BLaStM Trial	5 February 2015 and June 2028	Prostate Cancer	Interventional (Clinical Trial)	164 participants	Intervention Model: Parallel Assignment; Randomized	Treatment

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
