# Peer review of "Bête Noire of Chemotherapy and Targeted Therapy: CAF-Mediated Resistance"

_cancers, 2022, doi:10.3390/cancers14061519_

Round 1

Reviewer 1 Report

In the review article “Bête Noire of Chemotherapy And Targeted Therapy: CAF Resistance” the authors provide an overview of the current literature on cancer-associated fibroblasts (CAFs) and their role in the development of tumor therapy resistance with a focus on chemotherapy and targeted therapies.

Overall the review addresses an interesting and relevant topic, which is also reflected by a number of existing review articles covering the topic from different perspectives. This review focuses on chemotherapy and targeted therapies and provides an overview on current ongoing clinical trials.

While the authors provide a comprehensive overview of the current literature (151 references), the review is a little weak in providing novel insights or perspectives.

Specific considerations:

The authors write that “CAFs are "called into action"/activated by tumor cells” (lines 17 and 694). As cancer cells are no thinking beings that consciously call something into action the authors may want to discuss or speculate what other physiological scenarios (e.g. wound healing) are known where fibroblasts would respond to signals of epithelial (stem) cells by secretion of proliferation enhancing signals? Are the signaling pathways in these events the same or similar?

The authors need to pay close attention to precise wording. It starts with the title: “Bête Noire of Chemotherapy And Targeted Therapy: CAF Resistance.” It is not CAF resistance but rather CAF mediated resistance.

As CAF stands for cancer-associated fibroblast it should in most cases be used in plural (CAFs). Therefore, instead of “… CAF interacts …” (line 95) or “CAF orchestrates …” (line 185) it should be “CAFs interact” or “CAFs orchestrate”.

Line 240: “Condition media” should be “Conditioned media”.

The authors should check the entire text for consistency regarding the naming of genes and proteins. For example “neuregulin 1 (NRG1)” is correctly introduced in line 220 but subsequently used in different forms: “neuregulin 1” (line 220), “NRG1” (line 224), “neuregulin 1 NRG1” (line 287).

Some sentences are hard to understand or are missing words and should be revised. For example: “In lung adenocarcinomas, primary EGFR-TKI resistance is mediated via hepatocyte growth factor in CAF, which reportedly epidermal growth factor receptor tyrosine kinase inhibitor (EGFR-TKI) resistance in cancer cells bearing EGFR mutations” (line 289), “The established paracrine cross-talk between cancer cells and CAF regulated CAF physiology, which in turn imparted resistance to cancer cells” (line 266), or “Pancreatic tumors present a highly fibrotic stroma of activated CAFs surrounding the tumor cells and create an immune-suppressive TME via secretion of immunoregulatory and chemoattractive factors, preventing tumor-reactive T-cells responses occurs in addition to the resistance to anti-angiogenic targeted therapy and chemotherapy due to poor vascularization with poor perfusion” (line 411). The authors should check the entire text to identify and revise such sentences. “The role of CAF in several common solid tumors, as well as rare solid tumors, as presented above, bring out some interesting facts” (line 707). This sentence is another example for imprecise wording. It is not the role of CAFs that brings out facts. There are rather interesting facts (or better observations/properties) about the role of CAFs in tumor progression or tumor resistance to treatment.

Figure 1 / Line 109: “(organ types of tumors are indicated by their respective ribbon colors).” What color represents which tumor?

Line 112: “Figure 1” should be put into parentheses.

Line 120: “IL-11 secretion from CAF resisted chemotherapy regimens in gastric cancer cells targeting gp130/JAK/STAT3/Bcl pathway has been reported [27].” Please clarify: is it the secretion that resisted chemotherapy, or did IL-11 secretion from CAF lead to resistance or was associated with resistance?

Table 1 summarizes 17 ongoing and completed clinical trials involving CAF posted on ClinicalTrials.gov. However, the table was not included in the manuscript document and hence was not available for review. Starting from line 543 the authors describe the two completed studies out of all 17 studies. Just describing the purpose of the studies is of little value if no results or conclusions are presented. The same is true for the following description of ongoing trials. The only value of giving an overview of current trials is to underscore the general interest in and potential of therapeutically targeting CAFs.

However, the observation of a lack of basic and translational data regarding gynecological malignancies such as ovarian and more so in endometrial cancers (line 664) is a valuable statement to be discussed in a review article.

The authors should pay attention to the tense used. In the following example they first use past tense (tested) and then future tense (will investigate). “In contrast to the above studies, the Phase1/2 recruiting trial NCT04939610 (LuMIERE) tested 68Ga-FAP-2286 and 177Lu-FAP-2286 for the primary purpose of treatment. This Multicenter, Open-label, Non-randomized study will investigate the safety, tolerability, pharmacokinetics, dosimetry, and preliminary activity of 177Lu-FAP-2286 in 170 participating patients with advanced solid tumors” (line 632).

The first three paragraphs of the final section (“Forward Thinking”) consist of a repetition or summary of some parts of previous sections. Only the last paragraph provides a little – but still rather vague – outlook on future perspectives. Instead of saying “we will have to design a workable model to test the hypothesis” (line 736) the authors should provide a precise hypothesis and suggest a strategy to test the hypothesis.

In summary, the manuscript needs substantial editing regarding the precise wording of many sentences and the additional value of this review in comparison to other existing reviews should be made more clear. What is the take home message?

Author Response

Response to Comments from the Reviewer #1

We appreciate the comments from Reviewer #1. We agree with the reviewer's concerns. Our comment-by-comment response to all the comments/suggestions are presented below:

Reviewer's comment: The authors write that "CAFs are "called into action"/activated by tumor cells" (lines 17 and 694). As cancer cells are no thinking beings that consciously call something into action the authors may want to discuss or speculate what other physiological scenarios (e.g. wound healing) are known where fibroblasts would respond to signals of epithelial (stem) cells by secretion of proliferation enhancing signals? Are the signaling pathways in these events the same or similar?

Authors' Response: Line 17: As suggested, we edited the sentence which now read as the following: "Since cancer-associated fibroblasts are "called into action"/activated by tumor cells, cancer-associated fibroblasts provide an unconditional servitude to tumor cells in their effort to resist treatment" in the revised MS.

Line 694: As suggested, we removed the sentence.

Reviewer's comment: The authors need to pay close attention to precise wording. It starts with the title: "Bête Noire of Chemotherapy And Targeted Therapy: CAF Resistance." It is not CAF resistance but rather CAF mediated resistance.

Authors' Response: As suggested by the reviewer, we have changed the title of the MS to "Bête Noire of Chemotherapy And Targeted Therapy: CAF Mediated Resistance."

Reviewer's comment: As CAF stands for cancer-associated fibroblast it should in most cases be used in plural (CAFs). Therefore, instead of "… CAF interacts …" (line 95) or "CAF orchestrates …" (line 185) it should be "CAFs interact" or "CAFs orchestrate".

Authors' Response: Line 95: As suggested by the reviewer, we have edited the sentence.

Line 185: As suggested by the reviewer, we have edited the sentence. Similarly, we have edited the abstracts and the rest of the MS.

Reviewer's comment: Line 240: "Condition media" should be "Conditioned media".

Authors' Response: As suggested by the reviewer, we have edited the sentence.

Reviewer's comment: The authors should check the entire text for consistency regarding the naming of genes and proteins. For example, "neuregulin 1 (NRG1)" is correctly introduced in line 220 but subsequently used in different forms: "neuregulin 1" (line 220), "NRG1" (line 224), "neuregulin 1 NRG1" (line 287).

Authors' Response: As suggested by the reviewer, we have edited the sentence as follows.

" CAF-derived NRG1 promoted resistance in tumor cells through activation of HER3 involving NRG1/HER3 axis proving a paracrine mechanism of antiandrogen resistance in prostate cancer."

Reviewer's comment: Some sentences are hard to understand or are missing words and should be revised. For example: "In lung adenocarcinomas, primary EGFR-TKI resistance is mediated via hepatocyte growth factor in CAF, which reportedly epidermal growth factor receptor tyrosine kinase inhibitor (EGFR-TKI) resistance in cancer cells bearing EGFR mutations" (line 289), "The established paracrine crosstalk between cancer cells and CAF regulated CAF physiology, which in turn imparted resistance to cancer cells" (line 266), or "Pancreatic tumors present a highly fibrotic stroma of activated CAFs surrounding the tumor cells and create an immune-suppressive TME via secretion of immunoregulatory and chemoattractive factors, preventing tumor-reactive T-cells responses occurs in addition to the resistance to anti-angiogenic targeted therapy and chemotherapy due to poor vascularization with poor perfusion" (line 411). The authors should check the entire text to identify and revise such sentences. "The role of CAF in several common solid tumors, as well as rare solid tumors, as presented above, bring out some interesting facts" (line 707). This sentence is another example for imprecise wording. It is not the role of CAFs that brings out facts. There are rather interesting facts (or better observations/properties) about the role of CAFs in tumor progression or tumor resistance to treatment.

Authors' Response: We are sorry for the mistakes. As suggested, we have edited the MS.

line 266: We edited the sentence which now reads" The established paracrine crosstalk between CAFs and cancer cells regulated physiological characteristics of CAFs, which in turn imparted resistance to cancer cells."

line 289: In lung adenocarcinomas bearing EGFR mutations, primary EGFR-TKI resistance is mediated via hepatocyte growth factor from CAFs.

line 411: We have edited the sentence as below: "Pancreatic tumors present a highly fibrotic stroma containing activated CAFs which create an immunosuppressive TME. CAFs secrete immuno-regulatory and chemo-attractive factors, preventing tumor-reactive T-cells' responses".

We have also discussed subtypes of CAF in the different sections of the review as suggested by other reviewers.

line 707: We have edited the sentence as below:

"The role of CAFs in several common and rare tumors, as presented above, gives us an idea about their role in (1) tumor progression and/or (2) modes of development of resistance to treatment."

Reviewer's comment: Figure 1 / Line 109: "(organ types of tumors are indicated by their respective ribbon colors)." What color represents which tumor?

Authors' Response: We are sorry for the confusion. We have added the ribbon colors to the figure legends.

Head and neck cancer Color: White and burgundy; Kidney cancer Color: Orange; Breast cancer Color: Pink; Uterine or endometrial cancer Color: Peach; Liver cancer Color: Emerald green; Ovarian cancer Color: Teal; Prostate cancer Col-or: Light blue; Colon cancer Color: Dark blue; Stomach cancer Color: Periwinkle blue; Testicular cancer Color: Light purple; Pancreatic cancer Color: Purple; All cancers Color: Lavender; Brain cancer Color: Grey; Lung cancer Color: White or pearl; Bladder cancer Color: Blue, yellow, and purple; Skin cancer Color: Black.

We have added the colors of appropriate cancer ribbons in both figures 1 & 2.

Reviewer's comment: Line 112: "Figure 1" should be put into parentheses.

Authors' Response: We are sorry for the sloppiness. In the revised MS, the edited sentence reads as presented below:

"The extracellular vesicles, secretome, and autocrine/paracrine modes are much less involved in the manner of action (Figure 1)."

Reviewer's comment: Line 120: "IL-11 secretion from CAF resisted chemotherapy regimens in gastric cancer cells targeting gp130/JAK/STAT3/Bcl pathway has been reported [27]." Please clarify: is it the secretion that resisted chemotherapy, or did IL-11 secretion from CAF lead to resistance or was associated with resistance?

Authors' Response: We are sorry for the sloppiness. In the revised MS, the edited sentence reads as presented below:

"Secretion of IL-11 from CAFs activated the IL-11/IL-11R/gp130/JAK/STAT3/Bcl anti-apoptosis signaling pathway in gastric cancer cells. Thus CAF-derived IL-11 secretion caused resistance to chemotherapy regimens in gastric cancers". 

Reviewer's comment: Table 1 summarizes 17 ongoing and completed clinical trials involving CAF posted on ClinicalTrials.gov. However, the table was not included in the manuscript document and hence was not available for review. Starting from line 543 the authors describe the two completed studies out of all 17 studies. Just describing the purpose of the studies is of little value if no results or conclusions are presented. The same is true for the following description of ongoing trials. The only value of giving an overview of current trials is to underscore the general interest in and potential of therapeutically targeting CAFs.

Authors' Response: We had uploaded the table during our initial submission. We are again uploading an editable version of the table.

The purpose of making table-1 was to present the overall interest of physician-scientists in the clinical usefulness of the CAFs. We restricted our effort to "describing the purpose of the studies" in this review to impress the readers that physicians and scientists are beginning to find it worth exploring this avenue to manage the disease. We thank the reviewer for finding it relevant. It will be worth interrogating the results and purpose of each trial as suggested by the reviewer, which warrants a separate review article. However, details of the results from every 17 trials and their in-depth interrogation are beyond the scope and limits of this review. We have added the trial NCT # for each clinical trial for an interested reader.

Reviewer's comment: However, the observation of a lack of basic and translational data regarding gynecological malignancies such as ovarian and more so in endometrial cancers (line 664) is a valuable statement to be discussed in a review article.

Authors' Response: line 664: We have added the following paragraph as suggested b the reviewer:

"The conspicuous lack of the role of CAFs in the development of drug resistance in endometrial tumors can be explained by the lack of (1) absence of relevant data about the characterization of CAFs based on a drug resistance condition in the context of different pathological parameters, genomic alterations and outcome data, and (2) absence of a correct model system. It is understood that a bulk of endometrial cancers are detected early, and patients undergo surgical resection. A drug-resistance condition in the advanced or late stage in endometrial cancers is rarely presented where the tumor tissue can be accessed surgically. The characterization of endometrial CAFs and their presentation in the context of pathological parameters, genomic alterations, and outcome data in the future will pave the pathway for developing a model to test endometrial CAFs' function in a drug resistance scenario."

Reviewer's comment: The authors should pay attention to the tense used. In the following example they first use past tense (tested) and then future tense (will investigate). "In contrast to the above studies, the Phase1/2 recruiting trial NCT04939610 (LuMIERE) tested 68Ga-FAP-2286 and 177Lu-FAP-2286 for the primary purpose of treatment. This Multicenter, Open-label, Non-randomized study will investigate the safety, tolerability, pharmacokinetics, dosimetry, and preliminary activity of 177Lu-FAP-2286 in 170 participating patients with advanced solid tumors" (line 632).

Authors' Response: We are sorry for the confusion. We have edited the MS. We have used the "GRAMMARLY PROGRAM" (Microsoft OfficeGrammarly Inc. Version, 6.8.261) to edit the MS.

Reviewer's comment: The first three paragraphs of the final section ("Forward Thinking") consist of a repetition or summary of some parts of previous sections. Only the last paragraph provides a little – but still rather vague – outlook on future perspectives. Instead of saying "we will have to design a workable model to test the hypothesis" (line 736) the authors should provide a precise hypothesis and suggest a strategy to test the hypothesis.

Authors' Response: As suggested, we have reduced the section and commented on the "workable hypothesis in the take-home message section presented below:

Hence, the best way to develop a "workable hypothesis" of CAFs functions would be to generate experimental evidence strictly organ-specific. It is imperative to know the functions of specific signals from different CAF subtypes within TME of organ-type cancer(s). We can exploit the information for (1) targeting the pro-normalization signals from CAF while attenuating the pro-growth/progression and immuno-suppressive CAF signals and (2) identification of potential CAF markers to investigate the mechanisms underlying the role of CAFs in the TME.

Reviewer's comment: In summary, the manuscript needs substantial editing regarding the precise wording of many sentences and the additional value of this review in comparison to other existing reviews should be made more clear. What is the take home message?

Authors' Response: As suggested, we have added a take-home message as presented below:

The undeniable role of the subpopulation-specific function of CAFs in tumor growth, progression, and drug/immunotherapy resistance directly provides evidence for the therapeutically targetable role of CAFs. The concept of normalization of TME by targeting CAFs remains unmet. CAFs are heterogeneous and organ-type specific in origin, markers, and function. Hence, the best way to develop a "workable hypothesis" of CAFs functions would be to generate experimental evidence strictly organ-specific. It is imperative to know the functions of specific signals from different CAF subtypes within TME of organ-type cancer(s). We can exploit the information for (1) targeting the pro-normalization signals from CAF while attenuating the pro-growth/progression and immuno-suppressive CAF signals and (2) identification of potential CAF markers to investigate the mechanisms underlying the role of CAFs in the TME.

Reviewer 2 Report

In this review, De Pradip and colleagues, summarize how CAFs could mediate resistance to chemotherapy and targeted therapy, as well as how this CAF-mediated resistance could be overcome by targeting specifically CAFs. This review addresses an important question, is well documented, and falls at a time where there is an increasing interest in CAFs heterogeneity and functions in cancers. The review is organized around seven different parts.

Overall, the review is well documented, asks important questions. However, the review is sometimes difficult to follow, presenting some lack of clarity, some redundancy, and some paragraph that could be shortened. Moreover, I believe a general reorganization of the manuscript would help to improve the clarity of the overall manuscript.

I suggest below the different points that could help the authors to improve their manuscript:

-1) General organization of the review

            As mentioned before the review is separated into 7 parts. Some of these parts could be fused and authors should use sub-headers to better organize each part.

  • Part 2 and 3, both speak about Chemotherapy resistance. authors could organize only one part and address sub-parties/subheader in function on how CAFs resist chemotherapy, as proposed in figure 1. indeed this could be true for other parts of the manuscripts where authors propose in the first paragraph possible sub-parties but do not use it, as an example, they describe different mechanisms to targets CAFs, each different mechanisms could be associated with different sub parties and allow to fuse the part 5 (regulation of CAF function a therapeutic opportunity) and 6 (NIH Clinical Trial)
  • Also, the aforementioned part contains text and reference, that should be present in the CAF resistance to chemo or targeted therapy, and present redundancy.

2) Some parts of the manuscripts are unnecessarily long e.g.  Lane 660 to 685 is, the same for the two first paragraphs of FORWARD Thinking. The review is sometimes hard to follow and the authors add many references with no explanation or not in a sufficient manner. Many sentences could be shortened, or present repeats, increasing the difficulty to read the review and the lack of clarity

3) the introduction should describe more deeply the TME and its importance, focus a little more about CAFs origin, function and heterogeneity.

4) Speaking of CAF heterogeneity, this notion is almost absent from the review. An increasing number of studies are focused on of CAFs heterogeneity and question the impact of this heterogeneity on CAF resistance to therapy. Even if CAF heterogeneity is not the focus of this review, opening the discussion on this question could be more pregnant especially in the FORWARD Thinking part of the review. In addition, CAF heterogeneity could be associated with better outcomes, or response to therapy, in the review CAFs are often seen for their bad side but could be mentioned also the anti-tumoral/protective effect of CAFs.

5) The three figures of the manuscripts are not at the level of expectation for science they should be re-work.

6) minor comments: recent works also highlight how CAF drives resistance to radiotherapy; this could be included in their review.

Overall, the review is of great interest and improving its organization, improving the clarity and redundancy, would allow to increase reader attention and focus on the important question that should be addressed in the future.

Author Response

Response to Comments from the Reviewer #2

We appreciate the comments from Reviewer #2. We agree with the reviewer's concerns. Our comment-by-comment response to all the comments/suggestions are presented below:

Reviewer's comment: -1) General organization of the review

            As mentioned before the review is separated into 7 parts. Some of these parts could be fused and authors should use sub-headers to better organize each part.

Authors' Response: We have reorganized the review as suggested. Different parts are numbered, and sub-parts are too, as suggested.

Reviewer's comment: Part 2 and 3, both speak about Chemotherapy resistance. authors could organize only one part and address sub-parties/subheader in function on how CAFs resist chemotherapy, as proposed in figure 1. indeed this could be true for other parts of the manuscripts where authors propose in the first paragraph possible sub-parties but do not use it, as an example, they describe different mechanisms to targets CAFs, each different mechanisms could be associated with different sub parties and allow to fuse the part 5 (regulation of CAF function a therapeutic opportunity) and 6 (NIH Clinical Trial)

Authors' Response: Following the suggestion, we have combined parts 2 & 3 into one part and introduced subheaders. Similarly, we integrated parts 5 & 6 into one part and introduced subheaders as required in the revised MS.

Reviewer's comment: Also, the aforementioned part contains text and reference, that should be present in the CAF resistance to chemo or targeted therapy, and present redundancy.

Authors' Response: We have introduced organization in the MS by adding headers and subheaders as suggested.

Reviewer's comment: 2) Some parts of the manuscripts are unnecessarily long e.g.  Lane 660 to 685 is, the same for the two first paragraphs of FORWARD Thinking. The review is sometimes hard to follow and the authors add many references with no explanation or not in a sufficient manner. Many sentences could be shortened, or present repeats, increasing the difficulty to read the review and the lack of clarity

Authors' Response: We have removed text and references from several parts of the MS, including the forward-thinking. We also declined to add details on the individual clinical trials as suggested by other reviewers. However, we had to add details about iCAFs, myCAFs, and a take-home message as suggested by another reviewer.

Reviewer's comment: 3) the introduction should describe more deeply the TME and its importance, focus a little more about CAFs origin, function and heterogeneity.

Authors' Response: As suggested, we have referred to our two recent reviews describing CAFs origin, function, interaction(s) with tumor cells,  and heterogeneity as presented below:

"A detailed description of the origin(s), function(s), interaction(s) with tumor cells, and heterogeneity of CAFs are previously reviewed by us elsewhere (PMID: 34680395; PMID: 34502029)".

Reviewer's comment: 4) Speaking of CAF heterogeneity, this notion is almost absent from the review. An increasing number of studies are focused on of CAFs heterogeneity and question the impact of this heterogeneity on CAF resistance to therapy. Even if CAF heterogeneity is not the focus of this review, opening the discussion on this question could be more pregnant especially in the FORWARD Thinking part of the review. In addition, CAF heterogeneity could be associated with better outcomes, or response to therapy, in the review CAFs are often seen for their bad side but could be mentioned also the anti-tumoral/protective effect of CAFs.

Authors' Response: We wholeheartedly agree with the comment. As suggested, we have added the following text highlighting the idea that CAF's anti-tumor property should be promoted from the therapeutic point of view. We have added the following text in the revised MS:

"It has to be recognized that CAFs heterogeneity could be associated with better outcomes or response to therapy as opposed to CAF's pro-tumor actions. Bhattacharjee et al. demonstrated direct CAF-tumor interactions as a tumor-promoting mechanism, mediated by myCAF-secreted- hyaluronan and inflammatory iCAF-secreted-HGF (PMID: 33905375). Pro-tumorigenic effects in their study were opposed by myCAF-expressed type I collagen, which suppressed tumor growth by mechanically restraining tumor spread. Their study directly indicated that there is a scope for the therapeutic maneuvering of CAF-function in favor of the patient outcome by targeting specific signals for the tumor-promoting function of CAFs while promoting the myCAF-expressed type I collagen. This report, similar to other articles (PMID: 33930309), indicated the possibility of establishing therapeutically-targetable CAF subtype-specific mediators for the future treatment directed towards stromal-normalization of desmoplastic tumors."

Also, to emphasize on the topic, we have added the following texts in the "summary" and "take-home message" of the revised MS respectively:

……………." differential targeting of tumor-promoting CAF mediators while preserving the specifc anti-tumor functions like type I collagen may "normalize" stroma from tumor-promoting to tumor restricting phenotype (PMID:33905375).."

………" We can exploit the information for (1) targeting the pro-normalization signals from CAF while attenuating the pro-growth/progression and immuno-suppressive CAF signals, and (2) identification of potential CAF markers to investigate the mechanisms underlying the role of CAFs in the TME."

Reviewer's comment: 5) The three figures of the manuscripts are not at the level of expectation for science they should be re-work.

Authors' Response: We have revised the 3 figures as suggested.

Reviewer's comment: 6) minor comments: recent works also highlight how CAF drives resistance to radiotherapy; this could be included in their review.

Authors' Response: We had briefly touched on the topic of resistance to radiotherapy in the original MS as presented below:

"From the current literature, it is evident that exosomal miRNA predominantly mediates platinum-based chemotherapy resistance (cisplatin and oxaliplatin) with a few exceptions to tamoxifen resistance in the breast [37] and radioresistance in colorectal cancers [38];[39]."

Responding to the comments, we have now added the following text in the revised MS:

 "In the context of resistance to radiotherapy, CAFs are highly radio-resistant even at high doses of radiation. CAFs resist apoptosis signals following radiation and become senescent to produce a distinct combination of immunoregulatory molecules. Hence acquired radioresistance has been associated with CAFs' function (PMID: 30463848; PMID: 33637118). A recent minireview summarized findings on the interaction between CAF, ionizing radiation, and immune cells in the tumor microenvironment (PMID: 33207781). Targeting CAFs, regulatory T cells, and tumor-associated macrophages, in combination radio-immunotherapies have been reported to improve cancer treatment (PMID: 30766539)".

Reviewer 3 Report

I have read with tremendous enthusiasm the manuscript by Dey et al titled Bête Noire of Chemotherapy and Targeted Therapy: CAF Resistance.

In this review article, the authors provide comprehensive understanding of the different molecular mechanisms involved in tumor-stroma (with an emphasis on CAFs) co evolution across a wide spectrum of cancers that might actively contribute towards therapy resistance. Most therapy regimens currently in use in clinic rely on effectively targeting the tumor compartment but there is not much known about effectively treating/targeting the stromal microenvironment or activated CAFs. Thus, there is an urgent need for knowledge regarding CAF directed stromal re-programming/targeting agents to enhance long term patient outcomes. Overall, the review provided by the authors is of high clinical relevance and is a thorough research on potential strategies that may be harnessed to effectively target activated fibroblasts. It will further help improve the scientific quality of the review if the authors can incorporate/discuss a few of the following points:

Comment 1

myCAF/ iCAF and therapy resistance in the context of PDAC

Activation of iCAFs in PDAC by tumour-secreted IL1 activates LIF/JAK/STAT pathways, thereby promoting tumor growth in Biffi et al. (2019). iCAFs are considered to be inflammatory CAFs that promote tumor progression, and this pathway is closely related to iCAFs. Recent studies have therefore targeted this pathway. By treating IL1+ pancreatic cancer tumor cells with Anakinra, a IL-1R antagonist, and combining it with CAFs, researchers reduced Th2 levels and improved survival of pancreatic cancer in mice (Brunetto et al., 2019). Clinical trials based on IL-1R antagonists are currently underway (NCT02021422). According to Zhang et al. (Zhang et al., 2018), inhibiting IL-1B reduced pancreatic tumor fibrosis and reduced drug resistance in mice with KPC. As shown by Das et al. (2020), neutralizing IL-1B with antibodies significantly enhanced the therapeutic efficacy of PD-1 and increased tumor infiltration by CD8+ T cells in mice. Clinical studies using IL-1B inhibitors are ongoing (NCT04581343). 

CAFs that are inflammatory have low levels of SMA and high levels of IL-6. IL-6 is secreted by CAFs, which suppresses NK cell activity and promotes PDAC metastasis (Huang et al., 2019). PDAC patients with tumor-induced IL-6 are more likely to have a systemic metabolic stress response, which hinders anticancer immunotherapy. A recent study also showed that high levels of IL-6 were associated with reduced response to therapy (Neumann et al., 2018). Also, it was discovered that PSCs secreted IL-6, which transformed noninvasive pancreatic progenitor cells into invasive PDACs (Nagathihalli et al., 2016). 

According to Ligorio et al., 2019, CAF-secreted TGFB increases cell proliferation and invasion in PDAC cells. In a study published in 2019, Biffi et al. discovered that TGFB promotes fibroblast differentiation into myCAFs. Therefore, TGFB has also become a research target for treating PDAC.

It will be wonderful if the authors could include a brief discussion of the mentioned studies (myCAF and iCAF subtypes and effects on therapy outcomes) that were recently published (and also cite the relevant references).

It will help improve the scientific merit of the review if the authors could include a brief discussion of the mentioned studies that were recently published (and also cite the relevant references).

Comment 2

CAFs communicate with pancreatic tumor cells in a variety of ways, according to recent studies. Currently, CAF research involves the intervention of a subpopulation of CAFs rather than the complete inhibition of CAFs, and preliminary results have been achieved. For example, Meflin+ (Mizutani et al., 2019) CAFs inhibit differentiation into myCAFs and suppress pancreatic cancer development. By blocking NetG1, CAFs increased the survival of PDACs in vivo (Francescone et al., 2020). LRRC15 positive CAFs were associated with poor response to anti-PD-L1 treatment (Dominguez et al., 2020).The antibody dug conjugate against LRRC15 showed robust pre-clinical efficacy (Purcell et al, 2018). CAFs have been shown to secrete NRG1 and 7E3 (an antibody against NRG1) inhibited tumor growth and migration when co-cultured with CAFs, according to Ogier et al. (2018).

It will help improve the scientific merit and quality of the current review if the authors could briefly discuss these very recently published miscellaneous studies on therapy resistance and CAF subpopulations in tumor.

Author Response

Response to Comments from the Reviewer #3

We highly appreciate the comments from Reviewer #3. We are thrilled to receive encouraging comments from the reviewer. We enthusiastically agree with the reviewer's every concern. Our comment-by-comment response to all the comments/suggestions are presented below:

Reviewer's comment 1: myCAF/ iCAF and therapy resistance in the context of PDAC

Activation of iCAFs in PDAC by tumour-secreted IL1 activates LIF/JAK/STAT pathways, thereby promoting tumor growth in Biffi et al. (2019). iCAFs are considered to be inflammatory CAFs that promote tumor progression, and this pathway is closely related to iCAFs. Recent studies have therefore targeted this pathway. By treating IL1+ pancreatic cancer tumor cells with Anakinra, a IL-1R antagonist, and combining it with CAFs, researchers reduced Th2 levels and improved survival of pancreatic cancer in mice (Brunetto et al., 2019). Clinical trials based on IL-1R antagonists are currently underway (NCT02021422). According to Zhang et al. (Zhang et al., 2018), inhibiting IL-1B reduced pancreatic tumor fibrosis and reduced drug resistance in mice with KPC. As shown by Das et al. (2020), neutralizing IL-1B with antibodies significantly enhanced the therapeutic efficacy of PD-1 and increased tumor infiltration by CD8+ T cells in mice. Clinical studies using IL-1B inhibitors are ongoing (NCT04581343). 

CAFs that are inflammatory have low levels of SMA and high levels of IL-6. IL-6 is secreted by CAFs, which suppresses NK cell activity and promotes PDAC metastasis (Huang et al., 2019). PDAC patients with tumor-induced IL-6 are more likely to have a systemic metabolic stress response, which hinders anticancer immunotherapy. A recent study also showed that high levels of IL-6 were associated with reduced response to therapy (Neumann et al., 2018). Also, it was discovered that PSCs secreted IL-6, which transformed noninvasive pancreatic progenitor cells into invasive PDACs (Nagathihalli et al., 2016). 

According to Ligorio et al., 2019, CAF-secreted TGFB increases cell proliferation and invasion in PDAC cells. In a study published in 2019, Biffi et al. discovered that TGFB promotes fibroblast differentiation into myCAFs. Therefore, TGFB has also become a research target for treating PDAC.

It will be wonderful if the authors could include a brief discussion of the mentioned studies (myCAF and iCAF subtypes and effects on therapy outcomes) that were recently published (and also cite the relevant references).

It will help improve the scientific merit of the review if the authors could include a brief discussion of the mentioned studies that were recently published (and also cite the relevant references).

Authors' Response: As suggested, we have included the following text and associated references in the revised MS:

"Activation of iCAFs in PDAC by tumor-secreted IL1 activates LIF/JAK/STAT pathways, thereby promoting tumor growth in Biffi et al. (PMID: 30366930). iCAFs are considered to be inflammatory CAFs that promote tumor progression, and this pathway is closely related to iCAFs. Recent studies have therefore targeted this pathway. By treating IL1+ pancreatic cancer tumor cells with Anakinra, a IL-1R antagonist, and combining it with CAFs, researchers reduced Th2 levels and improved survival of pancreatic cancer in mice (PMID: 30760333 Brunetto et al., 2019). Clinical trials based on IL-1R antagonists are currently underway (NCT02021422). Neutralizing IL-1B with antibodies significantly enhanced the therapeutic efficacy of PD-1 and increased tumor infiltration by CD8+ T cells in mice (PMID: 31915130))).  Clinical studies using IL-1B inhibitors are ongoing (NCT04581343). CAFs that are inflammatory have low levels of SMA and high levels of IL-6. IL-6 is secreted by CAFs, which suppresses NK cell activity and promotes PDAC metastasis (PMID: 31598393)".

Reviewer's comment 2: CAFs communicate with pancreatic tumor cells in a variety of ways, according to recent studies. Currently, CAF research involves the intervention of a subpopulation of CAFs rather than the complete inhibition of CAFs, and preliminary results have been achieved. For example, Meflin+ (Mizutani et al., 2019) CAFs inhibit differentiation into myCAFs and suppress pancreatic cancer development. By blocking NetG1, CAFs increased the survival of PDACs in vivo (Francescone et al., 2020). LRRC15 positive CAFs were associated with poor response to anti-PD-L1 treatment (Dominguez et al., 2020).The antibody dug conjugate against LRRC15 showed robust pre-clinical efficacy (Purcell et al, 2018). CAFs have been shown to secrete NRG1 and 7E3 (an antibody against NRG1) inhibited tumor growth and migration when co-cultured with CAFs, according to Ogier et al. (2018).

It will help improve the scientific merit and quality of the current review if the authors could briefly discuss these very recently published miscellaneous studies on therapy resistance and CAF subpopulations in tumor.

Authors' Response: As suggested, we have included the following text and associated references in the revised MS:

"In the view of the finding that both cancer-promoting CAFs (pCAF) and cancer-restraining CAFs (rCAF) exit, CAF research currently involves the intervention of a sub-population of CAFs rather than the complete inhibition of CAFs. Meflin+ (PMID: 31439548..Mizutani et al., 2019) CAFs inhibit differentiation into myCAFs and suppress pancreatic cancer development indicating Meflin as a marker of rCAFs that suppress PDAC progression. "

As suggested, we have also added the following text describing "myCAF/ iCAF and therapy resistance in the context of PDAC and a few solid tumors" covering how CAFs communicate.

"CAFs are heterogeneous in terms of their origin in different organ-type cancers as well as the progression of the disease. The heterogeneous subpopulation of CAF such as myoblastic CAF (myCAF) and inflammatory CAF (iCAF), is extensively studied in fibroinflammatory PDAC disease characterized by dense and highly proliferating desmoplastic stroma. In fact, Li et al. have identified genes associated with the differentiation degree of myCAF and iCAF (PMID: 34140640; PMID:34948209; PMID: 32931156). Adipose-derived MSCs (AD-MSCs) have been shown to possess a high multilineage potential and self-renewal capacity and are reported as the CAF sources in PDAC by Miyazaki et al. (PMID: 32931156). Their study identified that AD-MSCs could differentiate into distinct CAF subtypes,  myCAF and iCAF, depending on the different co-culture conditions in vitro.

How diverse are the presence and functions of CAF subtypes? The diverse functions of iCAF and myCAF have also been reported in cholangiocarcinoma, breast cancers, prostate, head and neck squamous cell carcinoma, bladder and colon cancers. The diversity of CAFs subpopulations is also recently reported to promote the growth of cholangiocarcinoma, wherein hepatic stellate cells (HSC) are the primary source of CAF differentiating into myCAF and iCAF (PMID: 34129825). The hyaluronan synthase 2, but not type I collagen expressing myCAFs promoted tumor while HGF-expressing iCAFs enhanced tumor growth via tumor-expressed MET, thereby directly linking CAF to tumor cells. Another subset of CAFs, FAP+CAFs, is identified by Kieffer et al. in breast cancers that mediated immunosuppression and immunotherapy resistance via a positive feedback loop between specific CAF-S1 clusters and Tregs (PMID: 32434947). In prostate cancer, a differential mode of activation of iCAF and myCAF has been reported (PMID: 34520582). IL-1a/ELF3/YAP pathways are involved in iCAF differentiation, while TGF-beta1 induced myCAFs.

How do subtypes of CAFs interact with tumor cells? One of the ways CAFs classically interact with tumor cell EMT function has been reported by Goulet et al. in bladder cancer, where  IL-6 cytokine was found highly expressed in iCAFs, and its receptor IL-6R was found on RT4 bladder cancer cells (PMID:30744595). Perhaps the most intriguing functional heterogeneity of CAF is reported by Pan et al. in PDAC-CAF exhibited organ-specific metastatic potentials leading to different levels of heterogeneity of CAFs in different metastatic niches (PMID:34727952).

How do subtypes of CAF signal within TME? Several cell signaling pathways have been reported to be involved in the functioning of iCAF and myCAFs, including Hedgehog pathway (PMID: 33495315), Wnt-pathway (PMID:33055221), integrin a11B1 signaling (PMID:31159419), cMET-HGF pathway (PMID: 34129825), IL-6 signaling  (PMID:30744595),  EMT signaling via transcription factors SNAIL1, TWIST1 and ZEB1 (PMID:30744595), and IL1B-mediated crosstalk (PMID:34066976). Recently, Steele et al. reported that the Hedgehog pathway acts in a paracrine manner in PDAC, with ligands secreted by tumor cells signaling to stromal CAFs. The Hedgehog pathway activation is higher in PDPN+ alphaSMA+ myCAFs compared with iCAFs, and its inhibition impairs tumor growth by altering the fibroblast compartment in PDAC. Hedgehog pathway inhibition resulted in a reduction in myCAF numbers and a significant expansion of iCAFs, leading to an increase in the iCAF/myCAF ratio. As iCAFs are a source of inflammatory signals, authors observed an increase in iCAFs upon Hedgehog inhibition which correlated with changes in immune infiltration (significantly decreased CD8+ T cells and increased CD4+ T cells and CD25+CD4+ T cells; abundant FOXP3+ regulatory T cells) that are consistent with a more immunosuppressive pancreatic cancer microenvironment. The paracrine activation differentially elevated myCAF compared with iCAF leading to favorable alterations of cytotoxic T cells and Tregs, to cause increased immunosuppression (PMID: 33495315). Wnt-signaling in CAF represents a non-cell-autonomous mechanism for colon cancer progression (PMID:33055221). Mose et al. reported Sfrp1 induced epithelial-mesenchymal transition phenotype in tumor cells without affecting tumor-intrinsic Wnt signaling, suggesting an involvement of non-immune stromal cells. Low levels of Wnt signals induced iCAFs subtype which in co-culture with organoids induced EMT whereas high levels induced contractile myCAFs to attenuate EMT phenotype."

Round 2

Reviewer 1 Report

The authors have addressed all my comments and improved the manuscript. I have no further comments.

Reviewer 2 Report

No more comments, Authors reply to most of my concerns.

My only concern is about the figures that are still really difficult to read and not really informative.